# Exploring what is important during burn recovery: a qualitative study investigating priorities of patients and healthcare professionals over time

Christin Hoffmann ,[1,2] Philippa Davies,[3] Daisy Elliott ,[1,2] Amber Young [2,4]

The author sadly passed away on 17 September 2022. The paper was revised and resubmitted for publication by the corresponding author.

For numbered affiliations see end of article.

**Correspondence to**
Dr Christin Hoffmann;
c.hoffmann@bristol.ac.uk

## ABSTRACT

**Objectives** This qualitative study aimed to investigate: (1) priorities of patients and healthcare professionals during recovery from a burn injury, (2) how priorities change over time and (3) how priorities map to outcomes currently reported in burns research.

**Design** Semi-structured interviews were conducted. Interviews were audio recorded, transcribed and analysed thematically.

**Setting, participants** A total of 53 patients and healthcare professionals were recruited from four National Health Service (NHS) burn services across England and Wales across England and Wales. Patient participants (n=32) included adults, adolescents and parents of paediatric patients, with a variety of burn injuries in terms of severity and cause of burn injury. Healthcare professionals (n=21) were NHS staff members involved in burn care and included professionals with a range of clinical experience and roles (eg, nurses, surgeons, occupational therapists, physiotherapist, administration).

**Results** Ten themes relating to priorities (outcomes) during recovery from a burn injury were identified for patients and professionals. Of those, six were identified for patients and professionals ('pain and discomfort', 'psychological well-being', 'healing', 'scarring', 'function', 'infection'), three were unique to professionals ('patient knowledge, understanding and support', 'sense of control', 'survival') and one was unique to patients ('uncertainty'). Results highlighted that importance of these priorities changes over time (eg, 'survival' was only a concern in the short term). Likewise, priorities differed between patients and professionals (eg, 'pain' was important to patients throughout their recovery, but not for professionals). Seven out of 10 themes overlapped with outcomes commonly assessed in burn research.

**Conclusion** Professionals' and patients' priorities (important outcomes) change over time after burn injury and differ between those groups. Burn care research should consider measuring outcomes at different time points during the recovery from a burn injury to accurately reflect complexity of burn recovery.

## INTRODUCTION

Each year, approximately 10 000 patients with burn injuries are treated in the United Kingdom National Health Service (NHS)

⇒ A diverse sample of professionals and patients were recruited from four National Health Service burn services across England and Wales to elicit a wide range of perspectives.
⇒ The study employed semi-structured interviews with a large sample of 52 participants.
⇒ Multiple layers of analyses were applied, involving thematic analysis, comparison of views between patients and professionals and mapping themes to outcomes.
⇒ Identified themes are broadly relevant to patients with all types of burn injuries, but a more homogeneous sample of burn types may provide more detailed and nuanced insights into priorities of specific groups of patients.

burn services.[1] Burns can affect many aspects of patients' lives, including their function, cosmesis and psychological well-being. Injury management incurs challenges due to their varying complexity (eg, burn size and location) and factors unique to patients' personal circumstances (eg, age, profession).[2] Ongoing development of treatments to improve outcomes (eg, quality of life, scarring) means that evidence for optimal strategies for treating burns is still evolving.[3] Accurate measurement of patient-relevant outcomes to evaluate strategies is key to achieving high-quality, patient-centred care.[3 4]

The importance of relating patients' perceptions to measurable patient-relevant outcomes in burn research has been recognised.[5–7] Burns can impact affected individuals and their families in the short and in the long term in different ways.[8] For larger burns, for example, survival can be the main priority in the acute stages of treatment, while scarring may be a long-term concern. Similarly, scarring may be more or less important for some population groups and professionals involved their care.[9] It is probable

that perceptions about what matters most to patients and professionals may vary between those groups and over time. Consequently, the importance of outcomes may also vary. Current methods in burns research, however, commonly assume equal importance of outcomes at a specific time for all stakeholders.[10] Measuring a certain outcome at a certain time might not, however, provide a true picture of patients' recovery, and decisions about optimal treatment may, therefore, be ill informed. It is unclear how specific priorities differ across time as well as between patient and professionals as recovery from a burn injury progresses.

There is a need to identify which priorities are important at which time points during burns recovery. This may facilitate optimisation of outcome measurement and ensure that evaluation of treatment strategies is patient centred. Qualitative research is important in burns care[11] and has been widely used to explore perspectives and concepts important to burns survivors.[3 6 12] Previous work has shown that (mostly objective) clinical outcomes thought to be important, may not align with patients' views and perspectives.[13] A large number of qualitative studies have been conducted to explore concepts important to burns survivors only.[6] Likewise, outcome taxonomies have been proposed in this area and include patient-reported and patient-relevant concepts.[14 15] Less attention, however, has been paid to exploring professionals' views and literature often focuses on specific aspects of the recovery, such as scarring characteristics.[15] It is, therefore, important to understand professionals' *and* patients' priorities during burn recovery as both need to play a role in making decisions about treatment and care.[16 17] Likewise, understanding how they are linked to commonly measured outcomes in burn care research may guide future study design.

## Aim
This qualitative study had three objectives:
1. Investigate priorities of burns patients (including adults, adolescents and parents of paediatric patients) and healthcare professionals during burn recovery and explore how they differ.
2. Explore how priorities of patients and healthcare professionals change over time.
3. Map identified themes to outcomes currently reported in burns research.

## METHODS
This study uses qualitative methods, employing in-depth, semi-structured interviews to explore participants' perspectives over time.

Conduct and reporting of this study adhered to standards for reporting qualitative research.[18] This study adopted an interpretivist approach to understand participants perspective because it is acknowledged that experiences of all burn survivors are unique and that individuals' perceptions are construed within their own particular setting, which may implicate their families and social networks.[19]

## Setting and participants
This study recruited two groups of participants (patients and healthcare professionals) from four NHS burn service across England and Wales. Three were adult services and one specialised in paediatric burns. Patient participants were NHS patients with lived experience of burn injuries. These included adults (of 16 years and older), adolescents (10–15 years of age) and parents of paediatric patients (of less than ten years of age) (henceforth simplified to 'patients'). Patients with varying injury severity and burn sizes were eligible. Interviews were conducted at least 30 days after injury and/or with patients who were not in acute treatment (eg, larger burns). This ensured that all participants were able to reflect on their experiences around the time when the injury happened and during acute treatment and that they were also able to articulate what is most important to them when in the following stages of recovery. Healthcare professional participants were multidisciplinary NHS employees of burn services at the selected study sites (henceforth simplified to 'professionals').

Eligible participants were approached by research nurses or clinical/academic researchers and invited to participate in the study. Information about the study was provided and any questions were answered before written consent was obtained. A clinical (AY) or academic (CH) researcher, both trained and experienced in qualitative research, contacted participants to arrange a suitable interview method, location, date and time.

Purposive sampling was employed to select participants from a wide range of backgrounds.[20] Participant characteristics were reviewed regularly to ensure maximum possible variation between patients (eg, age, gender, severity and cause of burn injury) and professionals (eg, clinical experience, role). Recruitment continued until no new ideas and concepts evolved from the interviews (see 'analysis' section for further detail on our approach for data saturation).

## Data collection
Interviews with patients and professionals were completed face-to-face or over the phone, depending on participant preference. Face-to-face interviews were carried out in a quiet and private space in one of the four hospitals to ensure no interruptions or time limitations were impacting the interviews. In-depth interviews were conducted independently by two study members (AY and CH). AY is a consultant anaesthetist in paediatric burn care with 23 years of clinical experience and substantial experience of burns research. CH is a social scientist and qualitative researcher with no involvement in burns care or research prior to this study.

Interviews followed a semi-structured format using a topic guide with open-ended questions to guide discussions (see online supplemental tables S1 and S2 for examples).

Participants were asked to retrospectively reflect on their experience or priorities relating to different times after burn injury. There are no consensus definitions of time frames in burn care. To ensure consistency of explored perspectives, significant points during recovery from a burn injury were used as temporal anchors (eg, point of discharge is considered a crucial point for outcome assessment[21]). For the purpose of this study, time periods were, therefore, defined *a priori* as: short term (immediately after the injury), medium term (from around the time of discharge and/or wound healing) and long term (during rehabilitation). An initial topic guide was developed based on the study objectives of exploring priorities retrospectively over time. This was piloted prior to study commencement but allowed to evolve during the study, whereby iterative refinements were made as data collection progressed.

## Analysis

All interviews were audio-recorded using an encrypted device and deidentified during transcription. Random samples of verbatim transcripts were checked for accuracy by one researcher (CH).

Analysis of qualitative data was undertaken using a qualitative data management software (NVivo V.12). In a first step, transcripts were analysed using thematic analysis[22] to understand priorities of patients and professionals (study objective i). Interview transcripts were analysed separately for patients and professionals to accommodate concepts unique to each group and facilitate later examination of differences. This study used deductive and inductive approaches to coding.[23] Codes for short-term, medium-term and long-term priorities were developed *a priori*, to explore perspectives over time (study objective ii). *A priori* codes corresponded to time points (short term, medium term, long term) defined for the purpose of this study (described above). Additional codes were identified through line-by-line coding following principles of inductive theme development in line with six steps of thematic analysis.[22] Transcripts were read and reread to allow familiarisation with the data. Initial codes were assigned from which themes were developed: coded excerpts were grouped into themes of similar meaning and organised into a hierarchical structure of themes and subthemes. This thematic structure was refined as new themes emerged. These were then reviewed and refined through discussions among the study team until final definitions and labels were agreed: this process involved iterative rounds of interpretation of contexts specific to the interviewees' injury and identification of relationships between themes. Diagrams illustrating themes and relationships were used to visualise the data and aid discussions between team members. In a second step, differences in priorities between patients and professionals were examined by comparing narrative accounts of each group. Connections were drawn between patients and professionals by qualitatively contrasting themes identified for each group to find similarities and

differences in their perspectives (study objective i). This step also involved combining themes by condensing and relabelling which was shaped by the data and the team's experience and knowledge with defining, identifying and classifying outcomes in burns research.

Coding and analyses were undertaken independently by two researchers (CH and PD) who met regularly to discuss emerging themes from the data. Close contact between the coders during data analyses ensured consistency in coding and agreement of interpretation of the data. Independent double coding was also performed for 25% of transcripts by the two reviewing authors. A purposive sample of transcripts that underwent double coding was used to ensure maximum variation by participant group, sociodemographic background and injury severity. Data from double coding were compared and discussed to ensure consistency in approaches to further coding. Any uncertainties and discrepancies during this process were discussed with the wider study team and in consultation with an experienced qualitative researcher who was independent to the study team (DE). Interim interpretation and structure of themes were discussed with the wider study group and if necessary, iteratively modified.

Analyses were carried out in parallel to data collection to determine data saturation separately for both participant groups. This means, no new codes or meaning were identified through additional interviews and sufficient data were collected to address the research objectives.[24–27] Specifically, interim results during analyses were reviewed regularly by the study team and decisions of whether saturation was achieved were based on impressions of whether collected data provide sufficient conceptual depth.[28 29] If saturation was expected, a small number of additional interviews were conducted to confirm that no new codes or meaning were identified.[30] This process was overseen by the senior author (AY) who has a clinical background in burn care. A final thematic structure and meaning of themes were agreed in team discussions. Findings were collected in two descriptive accounts (one for patients, one for professionals) to be able to identify similarities and differences in key findings.

Themes identified through analyses described above were mapped against common outcomes in trials to assess whether perceptions explored in this work align with routinely measured outcomes in burns research (objective iii). This step involved comparing identified themes and their conceptual content to an existing outcome classification (see online supplemental table S3), which was developed through synthesis of 955 unique outcomes reported across 147 randomised controlled trials in burn care.[14]

This multilayer analysis allowed to elicit priorities qualitatively from experiences of patients and professionals while linking experiential data to previously identified and classified outcomes.

Participant characteristics were summarised using descriptive statistics. Data related to burn size were grouped into categories (minor, moderate, major)

according to percentage total body surface area, following established classification guidelines.[31]

## Patient and public involvement

This study is part of a wider project that aims to standardise outcome reporting and measurement in studies of burns injuries. Patient representatives and carers were actively involved in the design, conduct, analysis and dissemination of aspects of the wider project.

## RESULTS

### Participant characteristics

A total of 53 participants (32 patient participants, 21 professional participants) took part in semi-structured interviews, lasting between 15 min and 45 min (mean=32 min). Four patient interviews were conducted with two participants (three with a young person and their parents, one included a patient and their carer). Participants were recruited from four NHS trusts between June 2017 and February 2020.

Patient participants' age ranged from 1 to 82 years (mean=38; SD=27.4) and 19 (59%) participants were women. The sample mainly consisted of adults (n=20 participants, 65%). Scalds were the most common burn type (n=10, 36%). Participants primarily suffered from minor burns (n=14, 50%) with six (21%) patients who experienced major burns.

Healthcare professionals sampled were mostly women (n=17, 82%) with the majority of participants in a nursing role (n=11, 52%). Detailed participant characteristics are presented by participant group in table 1.

### Priorities of patients and professionals over time (objectives 1 and 2)

A total of 10 combined themes were identified as relating to participants' priorities during burn recovery. Six themes were most important for patients and professionals ('pain and discomfort', 'psychological wellbeing', 'healing', 'scarring', 'function' and 'infection') and four themes were identified as most important to only patients or professionals ('uncertainty', 'patient knowledge, understanding and support' 'sense of control' and 'survival'). There were differences in the relative importance of these priorities over time and differences between patients and professionals. An overview of combined themes and their importance for patients and professionals at different time points during recovery are found in figure 1. A detailed account of all themes identified for each stakeholder group and how each is linked to a combined theme and outcome domains can be found in online supplemental tables S1 and S2 of online supplemental file 1. The combined themes are summarised and presented in order of their prevalence in the data, ascertained by the number of coded references.

**Table 1** Participant characteristics (N=53)

| | Patients (N=32) | Professionals (N=21) |
|---|---|---|
| Age, mean (SD; range) | 38 (27.4; 1–82) | |
| Female, n (%) | 19 (59) | 17 (82) |
| Occupation, n (%) | | |
| Consultant clinical psychologist | | 3 (14) |
| Physiotherapist | | 1 (5) |
| Consultant, registrar | | 2 (10) |
| Occupational therapist | | 3 (14) |
| Research, specialist, clinic or senior nurse | | 11 (52) |
| Administration | | 1 (5) |
| Patient type, n (%) | | |
| Adult (16 years and older) | 20 (63) | |
| Young person (10–15 years) | 3 (9) | |
| Parent (of child aged<10 years) | 6 (19) | |
| Parent (of young person aged 10–15) | 3 (9) | |
| Burn size, n (%)* | | |
| Minor (<10% TBSA) | 14 (50) | |
| Moderate (10–20% TBSA) | 2 (7) | |
| Major (>20% TBSA) | 6 (21) | |
| Not reported | 6 (21) | |
| Burn reason, n (%)* | | |
| Scald | 10 (36) | |
| Flame | 8 (29) | |
| Chemical | 3 (11) | |
| Flash | 3 (11) | |
| Contact | 2 (7) | |
| Electric | 1 (4) | |
| Not reported | 1 (4) | |

*% data based on number of patient interviews (n=28) because this reflects the number of injuries
TBSA, total body surface area.

### Pain and discomfort (patients and professionals)

Pain was a concern common to patients and professionals in the short and medium term. Pain was a major cause of distress for the majority of patients and their family members as a result of the injury and initial treatment.

Professionals were particularly concerned about pain and adequate pain management immediately after the injury to prevent distress affecting patients in the long term. Reducing discomfort during wound healing, such

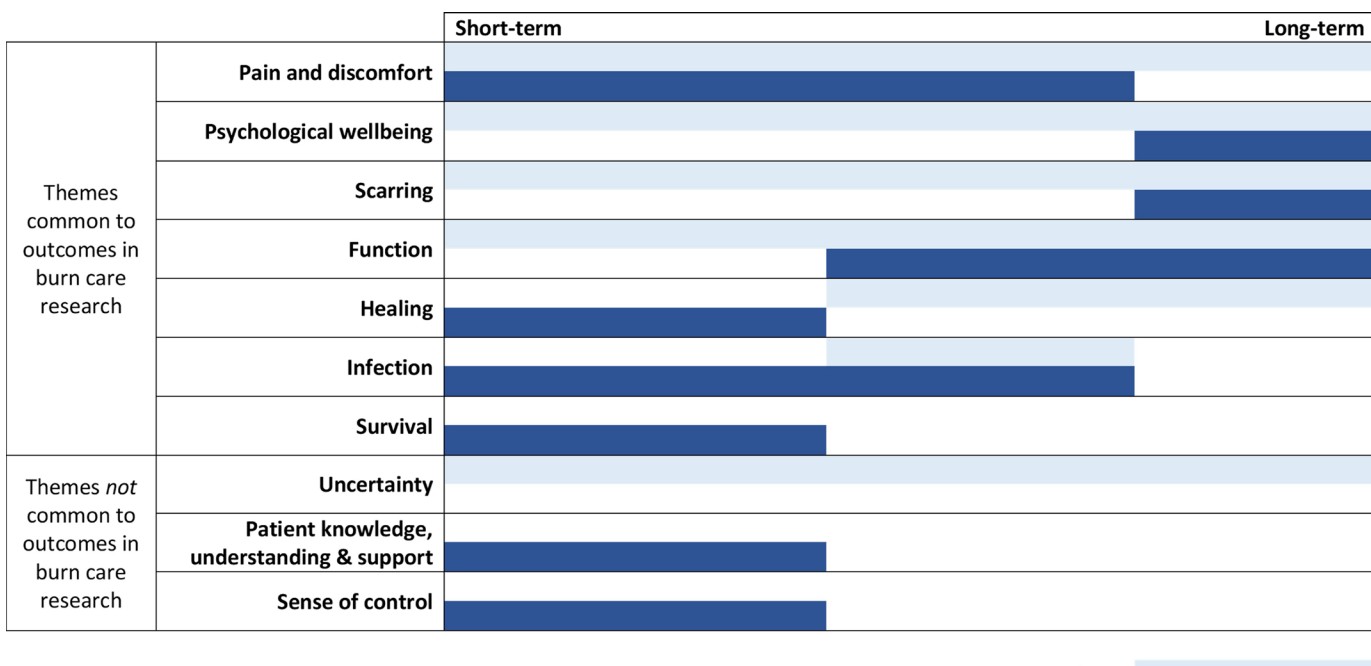

**Figure 1** Overview of patients' and professionals' priorities over time.

as easing itching and skin tightness were also raised as important in the medium term.

Some patients were worried about pain throughout their recovery. Especially, patients with larger burns, who complained about different aspects of long-term pain. For example, itching, scar tightness and contractures causing discomfort, neuropathic pain during nerve regeneration were all mentioned. Professionals, however, did not mention pain as a general concern in the long term.

Managing the pain is the immediate thing (professional, BCH30).

When it first happens it's just about the pain. That's it. […] that is just all you're focussing on at the time (patient, NBT012).

So the biggest problem is sometimes, you know, a bit of pain in the leg for example, but it doesn't stop me doing things, you know, like walking but sometimes […] it is perhaps because it's tighter than it should be, I don't know, but it was more painful but its easing (patient NBT007).

### Psychological well-being (patients and professionals)
Psychological well-being and related impacts were concerns common for patients and professionals in the long term. Both groups' concerns were predominantly linked to scar appearance, mentioning internal (eg, self-esteem, acceptance and normalisation of the injury) and external (eg, bullying, worry related to intimate relationships and other people's views) influences on psychological well-being.

Professionals were mainly worried about long lasting impacts on patients' psychological well-being, manifesting in post-traumatic stress disorder (PTSD) from distress caused by injury and scarring. In particular, patients with larger burns described psychological impacts during the later phases of recovery (eg, coping, PTSD).

For patients, the psychological effects of having a burn were apparent during all stages of recovery. In the short term, injuries affected individuals (eg, worry and anxiety) and family members (eg, anxiety and guilt, in particular, parents). Psychological impacts were also highlighted in the medium term by patients (eg, problems sleeping, flashbacks).

I mean long-term the important outcomes really are how they are psychologically (professional, BCH031).

When she saw her face she just broke down. She said 'I'm really ugly mummy' (patient, BCH001).

I was just getting too upset and depressed over the fact that I had burns and I looked so different and had too many worries and thoughts in my head that I just really didn't want to think about (patient, CW002).

### Healing (patients and professionals)
Healing of the wound was a common concern, expressed frequently by professionals and patients. Importance of healing across the stages of injury recovery, however, differed across participant groups.

Professionals worried about healing in the short term. This was evident through narratives about concerns relating to adequate management of the wound during the acute injury phase, including a quick and smooth process of assessment, cleaning and dressing.

Patients' worry about healing was apparent in the medium to long term of recovery and often linked to

feelings of surprise, frustration or worries about the amount of time it was taking for the wound to heal. The appearance of the wound was strongly linked to perceived length of healing times.

> Quite often we're very focused on dealing with the physical aspects of the burn injury, so making sure the burn wound heals, preventing infection. […] we're very focused on managing that wound and […] making sure the wound heals […] (professional, BCH014).

> When you see it it's quite gnarly, […] like when its fresh you're like oh god, this is never going to heal (patient, NBT002)

### Scarring (patients and professionals)

Scarring was repeatedly mentioned by both participant groups to be of importance in the long term.

Professionals were mostly concerned about appropriate scar management to optimise their appearance and avoid restrictions on movement.

Similarly, patients' worries in the later phases of recovery spanned any need for further treatment and potential impacts on movement as well as the appearance of scars influencing self-esteem and body image that could affect psychological well-being. For patients, the worries about risk of visible scarring were also evident in the short and medium term of the recovery. Those concerns affected individuals and their family members in terms of worry about their appearance and associated future impacts on the patient (eg, affecting psychological well-being).

> Long-term would be the scarring and whether contracture is affecting movement (professional, BCH013).

> Well the worries were the scarring and whether we'd need to have- at that point in time we didn't know- no-one could tell us how bad it was (patient, BCH001).

> Just wondering what the scarring would do, whether the skin would be very tight. Because you do see people who've had burns, like facial burns and sometimes their skin is very pink and red and then you do notice it. But yes, I think that was about it really. It's just how functional your hands would be when they're recovered (patient, NBT027).

### Function (patients and professionals)

Return to normal functioning was a concern common to patients and professionals in the medium and long term. Participants discussed worries about impacts of the injury (or scar) on daily activities and how it affected patients returning to their normal lives was an important concern following discharge.

Professionals' priorities in relation to function were also focused on aspects of mobility, primarily ensuring physical movement.

For patients, return to normal function was also a priority in the short term by highlighting concerns about the ability to return to work.

> The earlier you can intervene with physio then that's going to help with movement, […] it's important on-going as well if the wounds are that bad that there is movement issues (professional, BCH030).

> So initially I think it's kind of their return to their sort of family roles and their normal activities, when can I go back to work, when can I do my leisure activities (professional, NBT004).

> I was concerned with my hands to start off with because that is my livelihood (patient, NBT012).

> Particularly bathroom […], I absolutely hated the bedpans but I was too weak to get out of bed (patient, CW006).

### Infection (patients and professionals)

Infection was mentioned by patients and professionals. Importance of infection over time, however, differed across participant groups.

Preventing infection was one of the main priorities for professionals in the short and medium term. Following patients' hospital admission, professionals highlighted the importance of adequate cleaning and preparation of the wound to reduce healing time and scarring. Worry about cleanliness extended to the medium term of burn recovery when professionals wanted to ensure *that the patient understands how to look after their wounds, and to look out for signs of infection (professional, NBT026).*

Patients discussed worries about infection related to the medium term of their recovery. While most interviewed patients were aware of the need to keep the wound clean, infection was discussed as major concern primarily by those patients who had experienced symptoms of infection.

> Just to make sure there's no infection. If there's a lot of dead skin hanging around or blisters aren't de-roofed you could potentially get an infection just in the fluid that's still around or the dead skin (professional, BCH001).

> I thought that I had been released for good and was at home and actually I was extremely unwell. Night sweats, worrying about all sorts, just want, you know, really just completely in the dark about what was going to happen and I think because there's so much forewarning that it can happen, you know, then the worst does. It's a very upsetting thing to go through (patient, CW004)

### Patient knowledge, understanding and support (professionals only)

This theme was identified as a unique priority for professionals. In the medium term, at the point of discharge, professionals felt it was crucial to ensure patients are '*well equipped with the information that they need' (professional, BCH006).* This encompassed information related

to support the self-management of the wound (eg, creaming, massaging, sun protection). Professionals' main objectives were also to help patients understand their injury (eg, recognise signs of infection) and gain self-confidence in further wound treatment, so they remain as independent from the service as possible. Interviewees also repeatedly mentioned the importance of communicating the constant support available to patients, which included readiness to answer any questions and psychological support.

> It's explaining to them what they need to be doing to achieve this wound to heal (professional, NBT026).

> My biggest goal and concern for them is can they access everything they want to access (professional, BCH007).

### Uncertainty (patients only)

Uncertainty was identified as a unique theme for patients and was an important concern during the earlier stages of recovery. Feelings of uncertainty related to how the injury happened, the treatments needed and their possible outcomes as well as the likely course of recovery. Participants not only associated feelings of anxiety with uncertainty but also described how information provision (eg, about recovery and further treatment) can lessen those feelings. For patients with more severe burns, uncertainties remained in the long term and were mostly related to scar management and future treatment plans.

> During the time that we were kind of up in the air as to whether he would have the operation or not, we saw loads of different people on different rotas and stuff, and I kind of felt like there were lots of different opinions flying around about how well it was healing, how well it wasn't healing, what the next course of treatment would be. And although I understand why that was the case, because in the main they were kind of waiting to see what the wound was going to do, we felt very up in the air about […] what was going to happen, and whether what happened eventually would have actually been the right thing (patient, BCH017).

> Yeah, you've got no information, no knowledge and the pain materialises, you don't know whether that's supposed to happen, was that not supposed to happen, is something going wrong (patient, NBT012).

### Sense of control (professionals only)

For a smaller number of professionals, the sense of feeling in control of the treatment process was identified as a priority in the short term. This included major concerns about being in control of the treatment situation (eg, managing distress well) and technical steps involved. Professionals described the importance of feeling well prepared for the patients' treatment and the presence of a well-functioning team to support a smooth treatment. Consequences of lack of control were mentioned and

included impacts on self-confidence and patients' confidence in the care.

> It's being in control of that procedure. I feel as a clinical professional, […] once you've lost the control of the procedure in some way then you've lost the parents' confidence in you (professional, BCH006)

### Survival (professionals only)

Some professionals mentioned survival to be the key priority in the short term and were only discussed in the context of larger burns. For those professionals involved in their care, mortality-related outcomes and physiological signs (eg, organ failure, fever) were a major concern. Only a minority of patients experienced life-threatening injuries, but those with major burns generally did not discuss survival. This may be caused by *a kind of hole in their memory and they don't remember what happened to them (professional, MH006)*, aggravated by severe pain relief and reduced levels of consciousness.

> Well you'd be looking at their physical signs and outcomes, things like their vital signs, their bloods, whether they're being ventilated and they're on inotropes, and also whether they're having runs of sepsis and whether they've gone into acute renal failure (professional, NBT003).

> The most important outcome is survival because there are those patients that are very poorly and you just want them to survive (professional, BCH007).

### Mapping themes to outcomes (objective 3)

A total of seven out of 10 themes were identified as recognised outcomes in burn research. These were evident in a recently published systematic review that investigated all outcomes reported in burn care research[14] and identified research studies (see table 2).

There were three themes for which no outcome domains could be mapped. In particular, 'uncertainty', 'patient knowledge, understanding and support' and 'sense of control' were all themes uniquely identified in this study.[14]

## DISCUSSION

This study used semi-structured interviews to investigate perspectives of 53 burn patients, and professionals involved in their care, to explore what is most important as the patients recover. The study identified a total of 10 themes across both participant groups that related to priorities during recovery from a burn injury. Of those, six were identified across both participant groups ('pain and discomfort', 'psychological well-being', 'healing', 'scarring', 'function' and 'infection'), three were unique to professionals ('patient knowledge, understanding and support', 'sense of control', 'survival') and one was unique to patients ('uncertainty'). The study results also showed that the importance of recovery priorities

**Table 2** Comparison of identified themes and outcomes measured in trials

| Themes identified | Outcome domains mapped for both patients and professionals | Outcome domains mapped for patients only | Outcome domains mapped for professionals only |
|---|---|---|---|
| Pain and discomfort | ► Effect of scar on movement (contractures)<br>► Burn wound pain | ► Pain during procedures<br>► Itch<br>► Scar pain<br>► Comfort of dressings | ► Use of medicines to treat symptoms |
| Psychological well-being | ► Quality and quantity of sleep<br>► Psychological well-being<br>► Generalised anxiety<br>► Mental ability | | |
| Function | ► Ability to carry out daily tasks<br>► Return to work/school or previous function | ► Appearance | ► Mobility |
| Healing | ► Burn wound healing | ► Appearance | |
| Infection | ► Burn wound infection | ► Sepsis | |
| Survival | | ► Death from burn injury<br>► Death from any cause | |
| Uncertainty | None identified | | |
| Patient knowledge, understanding and support | None identified | | |
| Sense of control | None identified | | |

changes over time (eg, 'survival' was a concern in the short term but not in the medium to long term), and that priorities differed between patients and professionals (eg, 'pain and discomfort' was a concern in the short to medium term for professionals but important to many patients throughout their recovery). Finally, this work established that seven identified themes overlapped with outcomes commonly reported in burn research, whereas three themes could not be mapped to reported outcomes ('uncertainty', 'patient knowledge, understanding and support' and 'sense of control').

This work identified unique differences between what patients and professionals view as most important during burn recovery and supports the time sensitivity of priorities of patients and professionals. We were unable to find existing empirical evidence from burn care research that shows similar findings. This work, however, does complement other recent efforts to improve patient-centred outcome measurement in burn care research[3 6 7 12] by highlighting the need to align outcome assessment with the changing importance of priorities for different participant groups and with the change in relevance of priorities at different times after injury or intervention. Research in other healthcare areas has started to acknowledge these complexities of outcome assessment during recovery. Development of a core outcome set for childhood asthma, for example, reported the differences in perspectives of patients and professionals as to which outcomes are most important.[32] Other core outcome sets for lower limb amputation or induction of labour, for example, included separate outcomes relevant to short and medium-term recovery.[33–35] This study supports a recommendation for core outcome sets in burns care to differentiate between different time points,[3] but current core outcome sets do not commonly incorporate these complexities.[16]

An aim of on-going research is to ensure that frequently measured outcomes adequately capture all important priorities related to burns recovery.[6] The results from mapping themes to commonly measured outcomes in burn care research highlighted potentially important priorities currently not considered. Specifically, perceptions related to 'uncertainty' (patients), 'sense of control' and 'patient knowledge, understanding and support' (professionals) were all identified as important priorities at varying time points during recovery. The existence and impacts of uncertainty on patients are well known[36] and, as supported by our findings, have also been linked to the level of information provision.[37 38] Conversely, professionals in our study highlighted that 'patient knowledge, understanding and support' is one of their key priorities during the medium term of patients' recovery from a burn injury. Routine assessment of perceptions relevant to level of uncertainty and information provision as well as assessing patients' level of knowledge and understanding and support received can be important beyond commonly measured outcomes. In addition, these measures of subjective perception might provide useful care quality indicators to highlight where improvements, for example, in information provision, might be

required. Measurement instruments assessing patients' sense of control in healthcare interventions exist.[39 40] Very few studies, however, report the need to feel in control from the perspectives of healthcare professionals,[41] and we were unable to find a measure assessing professional specific sense of control. In our study, only a small number of professionals mentioned this as a priority, which warrants further research investigating the importance and impacts of perceived sense of control.

## Strengths and limitations

Our findings integrated views from a large sample of patients and healthcare professionals (n=53) from various backgrounds, allowing investigation of a variety of perspectives. This means that the findings are broadly representative of commonly occurring burn injuries but simultaneously may not account for unique concerns of specific patient groups. For instance, where burn injuries were related to mental health issues (eg, self-inflicted injuries), patients may display different patterns of priorities. Similarly, analysis focused on the most important and most commonly raised concerns, which implies that individual experiences and priorities may differ to themes identified in this study. Other limitations should be acknowledged. First, our sample consisted of patients with a variety of burn size which has implications for elicited perceptions across different time frames investigated. For instance, patients with larger burns experience longer healing.[42] This study defined time frames (short, medium and long term) *a priori,* which were agreed after expert consultation and discussions within the multidisciplinary study team. Individuals' definitions of short, medium and long-term recovery, however, are highly personal and might depend on contextual complexities (eg, size and location of burn, time since injury, socio-demographic details of patients, any wound complications) and might, therefore, differ to the definitions used in this study. Challenges to standardise time frames for recovery from burn injury are recognised considering the variabilities and complexities of injuries.[42–44] Long-term outcome measurement, for example, can range from a few months[45] to years[46] across studies assessing functional outcomes for large burns. Further research is warranted to pursue consensus definitions of time frames in burn recovery. In addition, time periods between participants' time of injury and interview varied across the sample, but no systematic variation in length of experience of recovery was pursued during recruitment. Due to resource limitations and practical considerations, the current work also did not employ a longitudinal study design but merely provides a snapshot of participants' perceptions. Quantitative and qualitative evidence suggests that varying recall periods may influence data collection in health research.[47 48] We were unable to collect reliable data on injury date to be able to assess the impact of potential recall bias caused by varying intervals on the results.

Second, our recruitment strategy meant that some patient groups may be underrepresented (eg, patients with larger burns, adolescent patients). A small number of qualitative studies showed that different population groups experience recovery differently. For example, women were better able to adapt to living with a burn injury[49] and older adults require more support and information[50] following severe burn injury. Future research could explore priorities in more homogeneous patient groups with larger sample size, to investigate whether these change between different populations and type of injury. Likewise, professional participants were mainly female and had nursing roles and there were challenges in recruiting surgeons, mainly due to individuals' time constraints. It is unclear if, for example, perspectives of male and female professionals differ and whether a more gender balanced sample would have impacted the results. There is value in exploring this difference in future work. It is also acknowledged that our sample lacks ethnic diversity. Again, it would be of value to explore perspectives over time specific to certain ethnic minority groups to reveal potentially unique and important culturally dependent priorities.[51 52]

Continued consensus work to refine existing core outcome and core measurement sets is crucial. Modular components of core sets using findings from this study could be developed, whereby different sets of outcomes are considered "core" to address priorities specific to time points and specific to different stakeholder groups. The use of applicable modules can accommodate differences in stakeholder priorities and optimise outcome measurement relevant to short-, medium- and long-term priorities. This would reduce research waste and outcome measurement burden, but requires effective strategies for implementation in practice.[53] Further research is also needed to better understand impacts of time sensitivity of outcomes during the recovery from burn injuries. This work provided a starting point to understanding the variability of the most important and broad priorities of patients and professionals during burn recovery but does not provide an outcome categorisation. Additional qualitative work might assess more granular aspects of themes identified in this work among more homogenous patient groups by drawing on existing outcome taxonomies.[6 7 14 54] This can include, for instance, exploring more nuanced psychological concepts that are important to specific patient groups (eg, adolescents or ethnic minority patients) or specific injury patterns (eg, large burns) over time. Likewise, priorities important across all stages of recovery require further detailed examination to be able to inform specific outcome measurement. Indeed, this work does not attempt to recommend how outcomes should be measured and further in-depth work is required for this. For example, different concepts related to pain (scar pain, burn wound pain, pain during the procedure) might be relevant at different time points. Further empirical work is necessary to explore commonly understood time frames for recovery periods (eg, short, medium, long term) which may facilitate the standardisation of time frames for outcome measurement in trials. This would allow efficient integration of essential and

meaningful outcome measures into trial design, avoiding research waste and enabling tailored treatment and intervention development in burns care.

## CONCLUSION

The study recommends that priorities (important outcomes) in burns care are measured at different time points during the recovery from a burn injury to accurately reflect complexity of burn care. In addition, it is suggested that different outcomes are considered for both patients and professionals. The inclusion of priorities not currently considered has the potential to uncover important areas for improvement of outcome assessment. This may ultimately benefit burn victims by more effective and improved patient-centred care. Further work is needed to elucidate nuances of priorities for specific patient groups and/or specific burn presentations and harmonise time periods within outcome measurement for efficient integration of time sensitivity of outcomes.

**Author affiliations**
[1]NIHR Bristol Biomedical Research Centre at University Hospitals Bristol and Weston NHS Foundation Trust and the University of Bristol, Bristol, UK
[2]Bristol Centre for Surgical Research, Population Health Sciences, Bristol Medical School, University of Bristol, Bristol, UK
[3]Population Health Sciences, Bristol Medical School, University of Bristol, Bristol, UK
[4]Children's Burns Research Centre Bristol, University Hospitals Bristol and Weston NHS Foundation Trust, Bristol, UK

**Acknowledgements** We would like to thank the research nurses Karen Coy, Catherine Spry and Anthony Sack for their help with recruitment for this study. We also thank all participants for dedicating their time and views to this study.Professor Amber Young, senior author, colleague and friend sadly passed away before the publication of this manuscript. Amber dedicated her career to advancing burn care research and worked tirelessly to improve outcomes for burn survivors. It was our privilege to work with Amber and to benefit from her skills, passion and knowledge. She will be truly missed.

**Contributors** AY conceived the idea for the project. CH wrote the manuscript with the support of AY, PD and DE. CH and AY performed data collection. CH and PD performed data analysis with support of AE and DE. DE is a qualitative researcher and provided oversight and guidance on methods. AY and DE edited and critically revised draft manuscripts. AY provided strategic oversight over the project and is responsible for the overall content as guarantor. All authors have read and approved the publication.

**Funding** This research is funded by The Scar Free Foundation, UK. The Scar Free Foundation is the only medical research charity focused on scarring with the mission to achieve scar free healing within a generation. The project was also supported by the National Institute for Health and Care Research (NIHR) Biomedical Research Centre (BRC) at the University Hospitals Bristol and Weston NHS Foundation Trust and the University of Bristol (BRC-1215-20011). AY has received an NIHR Doctoral Research Fellowship (DRF-2016-09-031 NIHR). This paper represents independent research funded by The Scar Free Foundation. The views expressed in this publication are those of the authors and do not necessarily reflect those of the Scar Free Foundation, the UK NHS or the NIHR.

**Competing interests** None declared.

**Patient and public involvement** Patients and/or the public were involved in the design, or conduct, or reporting, or dissemination plans of this research. Refer to the Methods section for further details.

**Patient consent for publication** Consent obtained directly from patient(s)

**Ethics approval** A favourable opinion for the project was granted by the South West—Frenchay Research Ethics Committee, reference 17/SW/0025. Participants gave informed consent to participate in the study before taking part.

**Provenance and peer review** Not commissioned; externally peer reviewed.

**Data availability statement** Anonymised transcripts are stored under 'controlled' access at the University of Bristol Research Data Storage Facility and any requests to access data will undergo committee review. Requests to obtain transcripts can be made to the corresponding author.

**ORCID iDs**
Christin Hoffmann http://orcid.org/0000-0002-6293-3813
Daisy Elliott http://orcid.org/0000-0001-8143-9549
Amber Young http://orcid.org/0000-0001-7205-492X

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
