## [Reviewer comments · BMJ Open]

ARTICLE DETAILS

TITLE (PROVISIONAL)	Exploring what is important during burn recovery: A qualitative study investigating priorities of patients and healthcare professionals over time
AUTHORS	Hoffmann, Christin; Davies, Philippa; Elliott, Daisy; Young, Amber

VERSION 1 – REVIEW

REVIEWER	Egberts, Marthe Utrecht University, Clinical Psychology
REVIEW RETURNED	10-Jan-2022

GENERAL COMMENTS	This article reports on priorities of patients and professionals during burn recovery. The manuscript is well written. A strength of the study is that both patients' and professionals' perspectives are taken into account and compared to each other. Relating the qualitative findings to burn outcomes assessed in research takes the study to a next level. However, the manuscript may be strengthened in several ways. Please find some suggestions below. Abstract • The objective is rather brief and does not include the aim of mapping identified themes to outcomes reported in burn research. Please make this more specific.• It would be helpful to include information on the timing of the interviews for patients, e.g., range in months postburn, since time-sensitivity is also mentioned to be an important aspect in the results. Introduction • The study objective/aim at p.4 and in the abstract is rather broad and vague, i.e., "what is most important". Do the authors refer to important burn outcomes, concerns, challenges, issues to be addressed in burn care? Please make this more specific. The information in the first paragraph under "methods" is already more explicit. To my opinion, this would better suit the "aims" section.• Please define the term "stakeholders" and "stakeholder group"; to whom do they refer? In general, this term is used throughout the article, but I think the term is rather distant/technical. Could the authors consider a more appropriate term? (or just use 'professionals' and 'patients')• "Current methods in burns research, however, commonly assume equal importance of outcomes at a specific time for all stakeholders." Can this be supported more extensively, including scientific references? Methods
---

	 • The supplemental topic guide for patients is helpful. It would be good to include the topic guide for interviews with professionals as well. • When was data saturation suspected? Was this determined for the patient and professional group separately? And was saturation checked with additional interviews? • The interviewers have different backgrounds. Were they both trained in qualitative interviewing? • Given the focus on change over time, it would be good to clarify that participants were asked to retrospectively reflect on different time periods, rather than being interviewed at multiple time points. • Overall, the analyses are described elaboratively. However, it would be helpful to include some more details on the method of inductive thematic analysis for study objective i. Which steps were taken in this thematic analysis? • I wonder which a priori codes were developed for study objective ii. What was the base for these codes? Can additional information be added? • Is there information available regarding response rates? How many participants were approached and how many consented? Results  • On p. 12 under “infection”, should the first line read: “Infection was mentioned by both participants and professionals”? • The description of the theme “sense of control” is rather short. At first, I thought it was about providing patients with a sense of control. As mentioned in the discussion, this is also a rather new theme. Can this theme therefore be elaborated upon? • It might be helpful to see at first glance which themes were identified for one specific group only, for example by noting this in the heading (e.g., Sense of control- professionals) • The addition of a simple figure displaying the themes in relation to the time frames might be helpful to obtain an impression of the results at first glance. Discussion  • There is heterogeneity in the group of recruited patients, related to age and burn severity, but also in role (patients vs. parents). Under “strengths and limitations” this is briefly mentioned, but this requires more attention. How does this influence the results? Can the authors integrate previous work on concerns/worries/outcomes for these different groups within the patient population? It would also be helpful if this heterogeneity in the population is referred to in the introduction, where it is stated that burns may affect individuals and families in different ways. • This study takes the qualitative findings to the next level by comparing them to outcomes in burn research, which is a strength. However, the fact that these outcomes are measure in burn research, does not mean that they are commonly monitored in clinical burn practice as well. It might be worthwhile to reflect on that in the discussion. • Under “Strengths and limitations”, it is mentioned that professional participants were mainly female and had nursing roles, but it is not stated how this may have influenced the results. Should other results be expected when more males and/or surgeons would have been recruited? • Under conclusion, it is not highlighted how patients and professionals may benefit from enhanced recovery assessment. Other
--	--

	 • Some references (e.g., #4) do not include author names but only initials.
--	---

REVIEWER	Woolard, Alix Telethon Kids Institute, Youth Mental Health
REVIEW RETURNED	22-Mar-2022

GENERAL COMMENTS	Thank you for allowing me to review this interesting and important manuscript titled 'Exploring what is important during burn recovery: A qualitative study investigating priorities of patients and healthcare professionals over time'. This paper aimed to investigate what patients and health workers deem important in burn recovery. I applaud the authors on the great sample size, 53 participants for a qualitative study is an impressive feat. The paper is well written and I only have minor comments that need to be addressed prior to publication. Minor comments  • In the introduction, there is background on the characteristics that might be important to patients, but I find there is a lack of background on the research done on health workers. A sentence or two would add to the introduction section and make the need for this paper clearer. • Why are the 16-18 year old participants included as adults? Typically paediatric services see up to 17-18 year olds. • Page 9 – should it say 'a young person'? • Can you include a standard deviation statistic for age in the demographic table? • Within the results, can you provide the number of patients/professionals who reported each theme being an issue? • Page 17 – I think this should say 'lower limb'
--

REVIEWER	Mathers, Jonathan University of Birmingham College of Medical and Dental Sciences, Institute of Applied Health Research
REVIEW RETURNED	23-Mar-2022

GENERAL COMMENTS	I think this paper would need a major revision to ensure that it (1) makes a significant and distinct contribution to knowledge that clearly adds to existing research in this field and (2) is demonstrably methodologically robust and (3) that the analysis presented is comprehensive and clearly organised into justified categories / themes / domains that are clearly devised and relate to existing domain frameworks that are already in the literature. I say this as an author of a couple of the frameworks referenced in the paper and with knowledge of other domain frameworks that have been proposed or devised alongside development of burn-specific quality of life tools. That could potentially make me somewhat biased in my view, but I actually thought that the most novel angle from this dataset might be a more detailed treatment of the perspectives of health care professionals, though perhaps that is a different paper. The key justification and focus of the paper seems to be the lack of clarity in existing frameworks as to the timing of relevant outcome priorities. I agree with this to an extent and I have just written a commentary piece focused on assessment of burn scar interventions that partly reflects on this. However, to an extent this
--

knowledge does exist, for example with identification of outcomes that are relevant to the acute / inpatient care existing in my own systematic review of adult qualitative research in burns. It is also debatable that we really do not know what are long term concerns for patients versus shorter term such as acute pain and survival. Finally, the conceptualisation here doesn't seem to consider the likely interrelationship between shorter term issues and things that persist e.g. acute trauma and issues influencing longer term psychological well-being and adjustment. There is existing research from sociology looking at this, specifically Janice Morse's work. There also doesn't seem to be a recognition of the potential tension in defining a priori time frames according to clinical consensus and then 'fitting' these to 'patient' narratives. I am sorry to be negative but as a sole justification for the paper this really needs strengthening and placing in existing knowledge.

Particularly, the depth of background and referencing would need to be placed to better and clearly develop a rationale for this paper and work. There are a number of existing frameworks (including my own), and some more recent work that has clearly defined the range of outcome domains – some from quality of life work e.g. construction of PRO tools, some simply derived from clinical perspectives, and some based on primary qualitative research.

Aims and approach

These are related comments to the above. Aim - "explore how priorities of patients and HCPs change over time" – this implies longitudinal research. But also priorities for what? This could be further explained. Further, under methods there are 3 points (?), objectives (?) aims (?) – how do they relate to single stated aim?

Methods

Interpretivism is referred to. I know this is difficult and complex but I did find this somewhat vague. Was the work informed by a specific qualitative methodological framework were you working within a generic qualitative framework. I think the latter is fine for this type of work and I would lay my own hat in that ring, but it does need recognising and acknowledging. How did previous qualitative research focused on patient / parent perspectives inform your approach? Please also see comments below about the methodological relationship between sampling, 'saturation', and inductive thematic approaches referenced – these are in my mind definitely in tension and strike me somewhat as 'checklistable' content as opposed to clear and justified methodological strategies.

Purposive sampling – there is just a single line on this. It would need more information about criteria to inform maximum variation sampling and particularly in relation to the longitudinal component of your research questions. How this was judged for adult patients, parents, adolescents, paediatric patients, and especially HCPs i.e. the diversity of the sample and what were you aiming for and why?

Saturation – you have mentioned recruitment until 'data saturation' but don't make any reference to methodological texts related to this to describe and justify this claim. There are a couple of issues with this. Firstly, you have not noted how you are making judgements of what you conceptualise as data saturation. For example, are the authors informed by work specific to judgments

of data completeness for outcome purposes, e.g. PROs and Cicely Kerr's work on this? Secondly, you later reference inductive thematic approaches to analysis, including Braun and Clarke from 2006. Braun and Clarke have specifically and recently challenged the use of data saturation conceptually in relation to their methodological work as it doesn't cohere with interpretivist qualitative approaches. Braun and Clarke would argue that it is a redundant concept in interpretivist research as you can't have a fixed reality to find in a dataset independent of researchers and analysts. Referencing their 2006 paper implies that the authors are not aware of this tension and does raise concerns about the internal coherency of the stated approach. This is difficult and complex stuff and I suspect data saturation is talked about because it is often expected, and in 'checklists', and Braun and Clarke 2006 being the go to methodological reference for a generic code based 'thematic' analysis. Whilst reported against a qualitative reporting checklist this seeming confusion of methods does raise concerns.

Coding – states that codes for short, medium and long term priorities were developed a priori. What were these codes? Or are you just saying you categorised according to your conceptualisation of short, medium and long term?

Sample

Table 1 – very little detail about participant characteristics, especially for 'patients' and especially considering you have very diverse interviews (parents, adolescents, adults).

There are only 5 parents in the sample? It's unclear. The table header says 32 but the detail adds up to 31 and then it says all parents = 5 and parents of children aged 10-15 is n=3 (part of the 5 and therefore total of 29?). There is not enough detail here. Clinical detail is not disaggregated by age etc. in the table. A reader cannot test your claims to a diverse purposive sample from the information presented.

Comparison of measures in burns trials

I have to confess to not understanding the purpose of this comparison. Why not compare to other conceptual work around outcomes and outcome domains in burns? The comparison provided seems to just give examples of measures of the 'themes' described. It doesn't provide a complete review of how the concepts have been measured and using what tools and whether that is comprehensive or not. As such I am not sure of the ultimate utility of this comparison. Why is there not a comparison to existing studies that outline outcome priorities / domains in burns. A comparison of conceptualisation and contents and justification for this. How is this similar or different? How does it add to existing knowledge? I think the paper really needs to do this if it is proposing a categorisation of things that important to patients as a way of justifying this to reflect on outcome assessment going forward.

As mentioned earlier I do think that the unique element in this paper is potentially the HCP interviews – perhaps as a standalone piece of work that might add to the literature.

Findings presented

These categories are very broad. Reading them and the contents I did think that this is only partially developed and again needs justification in relation to existing categorisations and frameworks. I

	was caused to question the thematic / category consistency. Two examples (1) Pain is reported as a most important priority but within this there is mention of itching, scar tightness, contractures and 'discomfort' – these are different to pain. (2) Psychological well-being – what is this and how is it defined and conceptualised? This is a really broad category and I'm afraid seemingly poorly described and conceptualised. Again this is difficult and complex but mentions PTSD, psychological well being, coping, worry, anxiety, guilt – again complex issues under a very simple category heading. Again I have to admit to a potentially skewed view here having spent much time myself analysing and organising conceptual categories in burns research. However, I am really concerned that a new categorisation, that doesn't refer or related to previous attempts, and is not that well justified, potentially clouds the water further. Priorities – you say most important here. What does that mean and how was it determined? The list of 'priorities' seems very limited compared to the range of issues we know are important to patients and parents. Again at risk of self-citation I would refer to the comprehensive list of outcome domains and items referenced in my own work and qualitative systematic review (spanning 41 qualitative studies). Discussion As mentioned above there is really no placement of this work within current knowledge and existing studies that present domain frameworks and categories. Limitations Were there any limitations to conducting these in hospitals. Personally I have often found time limited and context associated with clinical care and sometimes therefore discussion is stilted. These are quite short interviews – between 15 and 45 minutes – what implications does that have e.g. depth / richness. There doesn't seem to be any reflection on imitations of a non-longitudinal approach (sequential over time) or lack of specific sampling based on time point since injury / acute care? I am sorry if this review comes over as overly negative. I do think the novel angle here is the HCP interviews given time and effort to develop there perspectives on how working within and between services and supporting patients can affect outcomes. That seems novel considering existing knowledge.
--	---

VERSION 1 – AUTHOR RESPONSE

Reviewer: 1

Dr. Marthe Egberts, Utrecht University

Comments to the Author:

Comment 1.1

This article reports on priorities of patients and professionals during burn recovery. The manuscript is well written. A strength of the study is that both patients' and professionals' perspectives are taken into account and compared to each other. Relating the qualitative findings to burn outcomes assessed in research takes the study to a next level. However, the manuscript may be strengthened in several ways. Please find some suggestions below.

We thank the reviewer for the time to thoroughly review our manuscript and highlighting the strength of this work in interviewing both stakeholder groups, patients and professionals. We are grateful for suggestions to improve this manuscript, have addressed these below and implemented in the manuscript.

Comment 1.2

Abstract

- The objective is rather brief and does not include the aim of mapping identified themes to outcomes reported in burn research. Please make this more specific.

Thank you for spotting this omission in the abstract. We have included the objectives in the abstract on p.2, incorporating the reviewer's suggestions in comment 1.4.

The relevant section now reads: "Objectives: This qualitative study aimed to investigate: i) priorities of patients and healthcare professionals during recovery from a burn injury, ii) how priorities change over time, and ii) how priorities map to outcomes currently reported in burns research."

Comment 1.3

- It would be helpful to include information on the timing of the interviews for patients, e.g., range in months postburn, since time-sensitivity is also mentioned to be an important aspect in the results.

We entirely agree with the reviewer that this information would be valuable to report. We have, however, not been able to collect reliable data on the date of injury and can therefore only confirm that "All patients were more than 30 days after burn injury and were not in acute treatment" (p.5 of the manuscript). Narratives suggested that there has been a wide range of time points at which the injury happened, some of them several years. Such descriptions, however, are too unspecific to be reported as data points in this manuscript.

Comment 1.4

Introduction

- The study objective/aim at p.4 and in the abstract is rather broad and vague, i.e., "what is most important". Do the authors refer to important burn outcomes, concerns, challenges, issues to be addressed in burn care? Please make this more specific. The information in the first paragraph under "methods" is already more explicit. To my opinion, this would better suit the "aims" section.

Thank you for this helpful suggestion. We agree. We have made changes in line with the reviewer's recommendations and have removed any unclear and broad wording to list only the specific and clear objectives of this work. We decided for the word "priorities" as it best describes the focus of the

interviews from both, patient and professional, perspectives. Whilst we do mean “outcomes” in a scientific sense, we favour terminology that is inclusive of all stakeholder groups.

This resulted in changes to the abstract (p. 2) and added objectives to the “aim” section (pp. 4-5). The section was simplified to read:

“This qualitative study had three objectives:

- i. Investigate priorities of burns patients (including adults, adolescents and parents of paediatric patients) and healthcare professionals during burn recovery and explore how they differ.
- ii. Explore how priorities of patients and healthcare professionals change over time.
- iii. Map identified themes to outcomes currently reported in burns research.”

Comment 1.5

- Please define the term “stakeholders” and “stakeholder group”; to whom do they refer? In general, this term is used throughout the article, but I think the term is rather distant/technical. Could the authors consider a more appropriate term? (or just use ‘professionals’ and ‘patients’)

Thank you for highlighting this. We agree with the reviewer and have changed this throughout the manuscript where appropriate, to read “patients and professionals” or “participant groups”.

Comment 1.6

- “Current methods in burns research, however, commonly assume equal importance of outcomes at a specific time for all stakeholders.” Can this be supported more extensively, including scientific references?

Thank you for pointing out the need for a reference here. We have added a recent reference highlighting issues related to the lack of accounting for timing in outcome measurement in burn care research.

Mathers J. Towards the Holistic Assessment of Scar Management Interventions. Eur Burn J 2022, Vol 3, Pages 207-210 [doi: 10.3390/EBJ3010018]

We have also included an additional reference in the introduction which was very recently published (April 2022) which identified “changes over time” as an important overarching theme that explained that time was a vital aspect of recovery and the need for burn care to align with changing priorities over time.

Comment 1.7

Methods

- The supplemental topic guide for patients is helpful. It would be good to include the topic guide for interviews with professionals as well.

Many thanks for highlighting that the topic guide is helpful. As suggested, a second example topic guide for interviews with professionals has been added to the submission as Table S2 in a supplemental file.

Comment 1.8

- When was data saturation suspected? Was this determined for the patient and professional group separately? And was saturation checked with additional interviews?

Thank you for pointing out that further clarification is needed to describe how saturation and sample size were determined. No formal analysis for saturation was undertaken as we agree with current opinions about unsuitability of determining saturation a priori [1]. Instead, we chose to adopt approaches that focus on the saturation of both, codes and meaning and how sufficiently the data addresses the research objectives [2,3]. Therefore, determining saturation and the appropriate sample size is part of a qualitative judgment amongst a multi-disciplinary team which is inherently linked to findings emerging from data being collected. We have therefore added clarification in the analysis section on p. 6 of the manuscript to explain:

“Analyses were carried out in parallel to data collection to determine data saturation. This means, no new codes or meaning were identified in the interviews and sufficient data was collected to address the research objectives [2,3].“

In addition, a recent systematic review concluded that 9-17 interviews are necessary to reach saturation [4]. Our sample size exceeds guidance from this systematic review (32 patient participants, 21 professional participants), which provides us with increased confidence that sufficient data was collected.

Comment 1.9

- The interviewers have different backgrounds. Were they both trained in qualitative interviewing?

Thank you for raising this question and we can confirm that both interviewers are trained in qualitative methods, including doctoral-level qualifications. We have now included detail on p. 5 of the manuscript:

“A clinical (AY) or academic (CH) researcher, both trained and experienced in qualitative research, contacted participants to arrange a suitable interview method, location, date and time.”

Comment 1.10

- Given the focus on change over time, it would be good to clarify that participants were asked to retrospectively reflect on different time periods, rather than being interviewed at multiple time points.

Thank you for bringing this to our attention, we have added a sentence in the data collection section (p.6) to describe:

“Participants were asked to retrospectively reflect on their experience at different times after burn injury.”

It is worth adding that, in response to comment 3.2, we have also expanded on our rationale for including only patients that were 30 days post injury and not in acute care, i.e. to ensure that all

participants had transitioned to a long term recovery phase and were able to reflect on their short, medium and long term concerns.

Comment 1.11

- Overall, the analyses are described elaboratively. However, it would be helpful to include some more details on the method of inductive thematic analysis for study objective i. Which steps were taken in this thematic analysis?

Thank you for pointing out that our analysis section is elaborate. In view of this, we prioritised describing processes and steps in more detail that are less commonly used. We have now added a section on p. 6 to briefly describe our steps involved in applying thematic analysis which adhered to the recommended steps as described by Braun and Clarke [5]. The added section reads:

“Additional codes were identified through line-by-line coding following principles of inductive theme development in line with six steps of thematic analysis [5]. Transcripts were read and re-read to allow familiarisation with the data. Initial codes were assigned from which themes were developed: Coded excerpts were grouped into themes of similar meaning and organised into a hierarchical structure of themes and sub-themes. This thematic structure was refined as new themes emerged. These were then reviewed and refined through discussions amongst the study team until final definitions and labels were agreed: This process involved iterative rounds of interpretation of contexts specific to the interviewees’ injury and identification of relationships between themes. Diagrams illustrating themes and relationships were used to visualise the data and aid discussions between team members.”

Comment 1.12

- I wonder which a priori codes were developed for study objective ii. What was the base for these codes? Can additional information be added?

Thank you for raising this and we appreciate this requires further explanation. A priori codes were those that correspond to time periods used to structure the topic guide and somewhat standardise the different time points of burn recovery for the purpose of our study. These were described in the data collection section (p.6) and include: a) short term: immediately after the injury, b) medium term: from around the time of discharge and/or wound healing c) long term during rehabilitation.

We have added this information to the analysis section on p.6:

“*A priori* codes corresponded to previously defined time points (short term, medium term, long term).”

Comment 1.13

- Is there information available regarding response rates? How many participants were approached and how many consented?

Thank you for your interest in this data but unfortunately, this information was not collected during the study process. We adhered to SRQR guidelines in designing our research which did not specify reporting number of participants approached vs. consented. Most patients and professionals approached were happy to participate, but we will not be able to report this information reliably at this

occasion. It is worth noting, however, that bias in our sample might be possible due to the lack of ethnic diversity as detailed in our limitations section on pp. 17-18.

Comment 1.14

Results

- On p. 12 under “infection”, should the first line read: “Infection was mentioned by both participants and professionals”?

Thank you for spotting this mistake, this has now been corrected in line with the reviewer’s suggestion.

Comment 1.15

- The description of the theme “sense of control” is rather short. At first, I thought it was about providing patients with a sense of control. As mentioned in the discussion, this is also a rather new theme. Can this theme therefore be elaborated upon?

Thank you for suggesting that further descriptions are needed in this theme. We have changed the wording to clarify that this theme does not include providing patients with a sense of control. We have also expanded the paragraph to include further findings about perceived consequences of a lack of control. Changes were made on page 14 and now read:

“For a smaller number of professionals, the sense of feeling in control of the treatment process was identified as a priority in the short term. This included major concerns about being in control of the treatment situation (e.g. managing distress well) and technical steps involved. Professionals described the importance of feeling well prepared for the patients’ treatment and the presence of a well-functioning team to support a smooth treatment. Consequence of lack of control were mentioned and included impacts on self-confidence and patients’ confidence in the care.”

Comment 1.16

- It might be helpful to see at first glance which themes were identified for one specific group only, for example by noting this in the heading (e.g., Sense of control- professionals)

Thank you very much for this helpful suggestion. We have changed the sub-heading labels of the identified themes throughout the results section in line with the reviewer’s suggestion.

Comment 1.17

- The addition of a simple figure displaying the themes in relation to the time frames might be helpful to obtain an impression of the results at first glance.

Thank you for this suggestion. Figure 1 displays the themes in relation to the time frames by participant group which we believe addresses the reviewer’s comment.

Comment 1.18

Discussion

- There is heterogeneity in the group of recruited patients, related to age and burn severity, but also in role (patients vs. parents). Under “strengths and limitations” this is briefly mentioned, but this requires more attention. How does this influence the results? Can the authors integrate previous work on concerns/worries/outcomes for these different groups within the patient population?

It would also be helpful if this heterogeneity in the population is referred to in the introduction, where it is stated that burns may affect individuals and families in different ways.

We thank the reviewer for raising this interesting point. We have elaborated on this on p.18 to include qualitative evidence that suggests different patient population groups have different requirements and experiences. Most qualitative studies recruit only a small number of patients (8-20) and the true impact of age, gender and burn severity on outcome priority is not known from this body of evidence. There are larger epidemiological studies investigating impacts on physical outcome, e.g. patients experience more pain in older age [6]. To our knowledge, however, there is no work on sub-groups of patients with regards to outcome priority, which might be important for future research. We have highlighted this in our additions to the limitations section on p. 18.

“A small number of qualitative studies showed that different population groups experience recovery differently. For example, women were better able to adapt to living with a burn injury [7] and older adults require more support and information [8] following severe burn injury. Future research could explore priorities in more homogeneous patient groups with larger sample size, to investigate whether these change between different populations and type of injury.”

We have also followed the reviewer’s helpful suggestion and made minor changes to the introduction on p.4 to highlight examples of heterogeneity of experiences. It is important to point out, however, that this is the first study to investigate priorities over time and we aimed to sample a diverse group of population as a starting point for future investigations which may prefer to examine outcome priority over time for specific sub-groups highlighted in this study.

“Burns can impact affected individuals and their families in the short and in the long term in different ways [9]. For larger burns, for example, survival can be the main priority in the acute stages of treatment, whilst scarring may be a long term concern. Scarring may be more or less important for some population groups and professionals involved their care [10].”

Comment 1.19

- This study takes the qualitative findings to the next level by comparing them to outcomes in burn research, which is a strength. However, the fact that these outcomes are measure in burn research, does not mean that they are commonly monitored in clinical burn practice as well. It might be worthwhile to reflect on that in the discussion.

Thank you for supporting our study objectives and considering the outcome mapping activity a strength. The reviewer raises an important point which we fully endorse. This is a critical finding of this work which should be considered in future. The qualitative views reported here highlighted that some views were synonymous with outcomes commonly used and some priorities are not routinely measured. Standardising the measurement of outcomes across trials and clinical practice has received considerable research attention, specifically in burn care[11,12]. As such, core outcome sets have been proposed to optimise evaluation and address outcome heterogeneity in burn care. Facilitation of the uptake of core outcome sets is recognised to be an important area in need of

research and a crucial part of implementation science[13]. Both are beyond the scope of this work but have nevertheless caveated our discussion point on p. 19 to acknowledge such limitations “This would reduce research waste and outcome measurement burden, but requires effective strategies for implementation in practice [13].“

Comment 1.20

- Under “Strengths and limitations”, it is mentioned that professional participants were mainly female and had nursing roles, but it is not stated how this may have influenced the results. Should other results be expected when more males and/or surgeons would have been recruited?

Thank you and we agree that this is an interesting area but similarly to comment 1.18, it warrants further exploration in future work. Unfortunately, there is no published evidence that explores whether male or female professions have different views, so we cannot make certain claims in relation to this limitation. We have acknowledged this on p. 18 in the strengths and limitations section to say: “It is unclear if, for example, perspectives of male and female professionals differ and whether a more gender balanced sample would have impacted the results. There is value in exploring this difference in future work.”

Comment 1.21

- Under conclusion, it is not highlighted how patients and professionals may benefit from enhanced recovery assessment.

Thank you for highlighting this and we agree. We have now added the sentence “This may ultimately benefit burn victims by improving patient-centred care” on p. 18.

Comment 1.22

Other

- Some references (e.g., #4) do not include author names but only initials.

Thank you for noting the erroneous references. We have corrected this now.

Reviewer: 2

Dr. Alix Woolard, Telethon Kids Institute Comments to the Author:

Comment 2.1

Thank you for allowing me to review this interesting and important manuscript titled 'Exploring what is important during burn recovery: A qualitative study investigating priorities of patients and healthcare professionals over time'. This paper aimed to investigate what patients and health workers deem important in burn recovery. I applaud the authors on the great sample size, 53 participants for a qualitative study is an impressive feat. The paper is well written and I only have minor comments that need to be addressed prior to publication.

We thank the reviewer for their time and effort to review our manuscript and providing helpful comments which contributed to improving the work. We also appreciate the reviewer considering our paper "well written" and highlighting the large sample size.

Comment 2.2

Minor comments

- In the introduction, there is background on the characteristics that might be important to patients, but I find there is a lack of background on the research done on health workers. A sentence or two would add to the introduction section and make the need for this paper clearer.

Thank you for pointing this out and we agree that the introduction would benefit from more detail. Additions were made to the introduction on p. 4 and the last section now reads:

"A large number of qualitative studies have been conducted to explore concepts important to burns survivors only [14]. Less attention, however, has been paid to exploring professionals' views and literature often focusses on specific aspects of the recovery, such as scarring characteristics [15]. It is therefore important to understand professionals' *and* patients' priorities during burn recovery as both need to play a role in making decisions about treatment and care [11,16]."

It is worth noting that there is a paucity of qualitative research providing evidence on healthcare professionals' views in this area. This was also highlighted by the third reviewer as a strength of this work and valuable addition to the current body of knowledge.

Comment 2.3

- Why are the 16-18 year old participants included as adults? Typically paediatric services see up to 17-18 year olds.

The reviewer has highlighted a common discussion point. For instance, the National Institute for Care Excellence (NICE) in the UK defines their criteria up until the age of 18 in their guidance documents [17], supporting the reviewer's experiences. Most UK children services, however, only include patients up to 16 years of age and this definition is most widely adopted in the literature [18,19]. In any case, none of our participants were between the ages of 16-18, so a change in age definitions would not alter the data collected in this study.

Comment 2.4

- Page 9 – should it say ‘a young person’?

Thank you very much for spotting this mistake, this has now been corrected.

Comment 2.5

- Can you include a standard deviation statistic for age in the demographic table?

Thank you. We can confirm the SD of 27.4 is now included in the main body and table on page 8.

Comment 2.6

- Within the results, can you provide the number of patients/professionals who reported each theme being an issue?

We appreciate the reviewer’s request for including numerical data for each theme. We acknowledge that this approach may be suitable for some qualitative work (e.g. content analysis), but feel this is not adequate to include in our study. We have followed an inductive approach to analysis as described by Braun and Clarke [5], with iterative rounds of refinements of themes and sub-themes. This means that data generated is fluid and flexible, which might result in different responses whereby a number does not necessarily signify what all participants felt about a specific issue. Ultimately, a final number of references/quotes coded to each theme would not reflect the strength of evidence. These views are echoed by Braun and Clarke who discourage the use of frequency counts in reporting analysis, as ‘frequency does not determine value’ (p. 261) [20]. We have used language in our manuscript to indicate some elements of frequency, such as “many participants” or “some participants” which is better suited to show patterns within the data [5].

Comment 2.7

- Page 17 – I think this should say ‘lower limb’

Again, thank you for noticing this. This is now corrected.

Reviewer: 3

Dr. Jonathan Mathers, University of Birmingham College of Medical and Dental Sciences
Comments to the Author:

Comment 3.1

I think this paper would need a major revision to ensure that it (1) makes a significant and distinct contribution to knowledge that clearly adds to existing research in this field and (2) is demonstrably methodologically robust and (3) that the analysis presented is comprehensive and clearly organised into justified categories / themes / domains that are clearly devised and relate to existing domain frameworks that are already in the literature.

I say this as an author of a couple of the frameworks referenced in the paper and with knowledge of other domain frameworks that have been proposed or devised alongside development of burn-specific quality of life tools. That could potentially make me somewhat biased in my view, but I actually thought that the most novel angle from this dataset might be a more detailed treatment of the perspectives of health care professionals, though perhaps that is a different paper.

The key justification and focus of the paper seems to be the lack of clarity in existing frameworks as to the timing of relevant outcome priorities. I agree with this to an extent and I have just written a commentary piece focused on assessment of burn scar interventions that partly reflects on this. However, to an extent this knowledge does exist, for example with identification of outcomes that are relevant to the acute / inpatient care existing in my own systematic review of adult qualitative research in burns. It is also debatable that we really do not know what are long term concerns for patients versus shorter term such as acute pain and survival.

Finally, the conceptualisation here doesn't seem to consider the likely interrelationship between shorter term issues and things that persist e.g. acute trauma and issues influencing longer term psychological well-being and adjustment. There is existing research from sociology looking at this, specifically Janice Morse's work. There also doesn't seem to be a recognition of the potential tension in defining a priori time frames according to clinical consensus and then 'fitting' these to 'patient' narratives. I am sorry to be negative but as a sole justification for the paper this really needs strengthening and placing in existing knowledge.

Particularly, the depth of background and referencing would need to be place the work better and clearly develop a rationale for this paper and work. There are a number of existing frameworks (including my own), and some more recent work that has clearly defined the range of outcome domains – some from quality of life work e.g. construction of PRO tools , some simply derived from clinical perspectives, and some based on primary qualitative research.

We thank the reviewer for the thorough review of our manuscript and the time taken to provide comments which have significantly improved our manuscript.

It must be pointed out that the aim of this work was not to develop a new outcome domain framework or taxonomy and does not wish to endorse a particular framework. We have now stated this explicitly now on p. 19 to avoid confusion.

The contribution to knowledge of this study is that it showed that priorities differ over time and differ between burn survivors/patients and professionals (as illustrated in Figure 1). What is not new in our current study and not part of our investigation is exploring the depth of themes and outcomes and concepts related to each of those. The present research did not set out to comprehensively explore concepts and nuances within suggested themes/outcomes because, we fully agree with the reviewer, this has been the focus of extensive previous work and would greatly exceed the scope of this project. We also understand that there is existing literature on outcome domains in burn care [14,21,22] and applaud the extensive work that has been undertaken to progress this area of research. In fact, one of the co-authors contributed to this body of knowledge [22,23] and have indeed applied this knowledge in our analysis. This was now made more explicit in the methods section on p. 8 to explicitly state that we drew on this previous work "Existing outcome frameworks which comprehensively summarise outcomes in burn care were also cross-referenced [14,21,22]". We are conscious of not repeating or overruling any of this work and whilst we are aware of various frameworks for categorising outcomes,

the aim of the study was not to fit our results into any such framework but simply to draw on existing knowledge to contextualise our findings and to illustrate the change in priorities over time. We also did not suggest that the extracted concepts can be used as a standalone conceptual model or could serve as basis for determining measurable indicators of the many complex concepts of recovery from burns.

We have added content in the introduction and discussion to make reference to such frameworks and clarify what the study did NOT aim to do and that it merely serves as a starting point. As such, we have added on p.4: “A large number of qualitative studies have been conducted to explore concepts important to burns survivors only [14]. Likewise, outcome taxonomies have been proposed in this area and include patient-reported and patient-relevant concepts [15,22]. Less attention, however, has been paid to exploring professionals’ views and literature often focusses on specific aspects of the recovery, such as scarring characteristics [15].“ and on p. 19: “This work provided a starting point to understanding the variability of the most important and broad priorities of patients and professionals during burn recovery but does not provide an outcome categorisation. Additional qualitative work might assess more granular aspects of themes identified in this work amongst more homogenous patient groups by drawing on existing outcome taxonomies [14,21,22,24].”

Comment 3.2

Aims and approach

These are related comments to the above. Aim - “explore how priorities of patients and HCPs change over time” – this implies longitudinal research. But also priorities for what? This could be further explained. Further, under methods there are 3 points (?), objectives (?), aims (?) – how do they relate to single stated aim?

Thank you for highlighting that the use of the word “priorities” was not clear which was also raised by another reviewer. We acknowledge that clarification was needed, and as a result, significant changes were made in response to Comment 1.4.

We decided on a common minimum denominator for our inclusion criterion (i.e. at least 30 days post injury and not in acute treatment) at which point patients were asked to recall their experiences relevant to our pre-defined time points (clarified in response to comment 1.10). All patients would have transitioned to the “long term” recovery phase by then. We have expanded on our reasons on p.5 by including: “All patients were more than 30 days after burn injury and were not in acute treatment to ensure that participants could be considered within their long term recovery phase and were all able to reflect on their journey up until that point.”

Unfortunately, it is not feasible to interview patients at the time points of interest for this study (e.g. asking patients about their concerns directly after the injury happened), which means the use of longitudinal data collection was not possible. Likewise, some of our participants with larger burns may require several months until transition to long term recovery which implies following participants over a long time and extending the study length. We believe that such significant resource implications are warranted because we felt that patients were very able to communicate what was most important to them at the time, because they had time to reflect and overcome initial trauma that may have impacted their experiences.

Comment 3.3

Methods

Interpretivism is referred to. I know this is difficult and complex but I did find this somewhat vague. Was the work informed by a specific qualitative methodological framework were you working within a generic qualitative framework. I think the latter is fine for this type of work and I would lay my own hat in that ring, but it does need recognising and acknowledging. How did previous qualitative research focused on patient / parent perspectives inform your approach? Please also see comments below about the methodological relationship between sampling, 'saturation', and inductive thematic approaches referenced – these are in my mind definitely in tension and strike me somewhat as 'checklistable' content as opposed to clear and justified methodological strategies.

Many thanks for highlighting this. We have changed the relevant sentence on p.5 to explain why our stance aligns with the research context by explaining “This study adopted an interpretivist approach to understand participants perspective, because it is acknowledged that experiences of all burn survivors are unique and that individuals' perceptions are construed within their own particular setting which may implicate their families and social networks [25].”

We have also made additions to the analysis section on pp. 6-7 in response to comment 1.11 to explain in more detail how individual experiences were accounted for in the analysis.

We have indeed followed SRQR reporting guidelines as recommended by the journal and by adhering to this checklist, we have reported the underlying research paradigm. We felt that further information would distract from the content of our work and have included a reference for interested readers that prefer more detail than what current space limitations allow.

Comment 3.4

Purposive sampling – there is just a single line on this. It would need more information about criteria to inform maximum variation sampling and particularly in relation to the longitudinal component of your research questions. How this was judged for adult patients, parents, adolescents, paediatric patients, and especially HCPs i.e. the diversity of the sample and what were you aiming for and why?

Similar to our response to comment 1.8 where we explained why we consider pre-defined sample sizes inappropriate for qualitative research, a pre-defined list of criteria that participants need to meet, would equally be in appropriate and impractical to adhere to. This study was interested in perspectives from a wide range of patients and professionals with varying background. We were able to achieve this by applying one of the most commonly used technique of purposive sampling which is recommended since it is more likely to yield a greater variety of subjective lived experiences[26]. With limited space for detail, we prioritised describing other sections in greater detail. We have included the above reference for interested readers and have made it clearer that we mean “maximum possible variation” (p.5) rather than a pre-defined criteria and modified the section below which included the key criteria that were reviewed when deciding upon participant selection:

Purposive sampling was employed to select participants with from a wide range of backgrounds [26]. Participant characteristics were reviewed regularly to ensure maximum possible variation between patients (e.g. age, gender, severity and cause of burn injury) and professionals (e.g. clinical experience, role).”

Comment 3.5

Saturation – you have mentioned recruitment until 'data saturation' but don't make any reference to methodological texts related to this to describe and justify this claim. There are a couple of issues with this. Firstly, you have not noted how you are making judgements of what you conceptualise as

data saturation. For example, are the authors informed by work specific to judgments of data completeness for outcome purposes, e.g. PROs and Cicely Kerr's work on this? Secondly, you later reference inductive thematic approaches to analysis, including Braun and Clarke from 2006. Braun and Clarke have specifically and recently challenged the use of data saturation conceptually in relation to their methodological work as it doesn't cohere with interpretivist qualitative approaches. Braun and Clarke would argue that it is a redundant concept in interpretivist research as you can't have a fixed reality to find in a dataset independent of researchers and analysts. Referencing their 2006 paper implies that the authors are not aware of this tension and does raise concerns about the internal coherency of the stated approach. This is difficult and complex stuff and I suspect data saturation is talked about because it is often expected, and in 'checklists', and Braun and Clarke 2006 being the go to methodological reference for a generic code based 'thematic' analysis. Whilst reported against a qualitative reporting checklist this seeming confusion of methods does raise concerns.

Thank you for the detailed comment about saturation. We are aware of the ongoing discourse about use of the word saturation. The views of Braun and Clarke represent one perspective on this matter. In the absence of a clear consensus agreement, most recent qualitative literature (also published in BMJ Open) continues to use the term saturation [27,28]. We agree with the notion of providing transparency in how sampling was decided and prefer the approach of detailed descriptions of the meaning of "saturation" rather than refraining from using familiar terminology. Our response to Comment 1.8 elaborates on our justification for using this term and we have also made changes to the manuscript to detail what exactly was meant by saturation.

Further, we have specifically referenced the Braun and Clarke paper in relation to our method of analysis. We are fully aware of the more recent work by Braun and Clarke but did not feel that this is needed as we applied fundamental steps of thematic analysis published in the 2006 paper, which remains the dominant reference for thematic analysis in the literature. We provided further detailed on the method and steps undertaken in the methods section on p. 6, in response to Comment 1.11.

We would like to highlight again that this work is not meant to develop a new definition and conceptualisation of an outcome and feel that detailing work about completeness of outcomes is outside the scope of this study.

Comment 3.6

Coding – states that codes for short, medium and long term priorities were developed a priori. What were these codes? Or are you just saying you categorised according to your conceptualisation of short, medium and long term?

Thank you for raising this and we agree that further clarification was needed. Reviewer 1 has also made this observation, and this was addressed as part of our response to Comment 1.12.

Comment 3.7

Sample

Table 1 – very little detail about participant characteristics, especially for 'patients' and especially considering you have very diverse interviews (parents, adolescents, adults).

There are only 5 parents in the sample? It's unclear. The table header says 32 but the detail adds up to 31 and then it says all parents = 5 and parents of children aged 10-15 is n=3 (part of the 5 and

therefore total of 29?). There is not enough detail here. Clinical detail is not disaggregated by age etc. in the table. A reader cannot test your claims to a diverse purposive sample from the information presented.

Thank you for bringing this to our attention and we acknowledge that reporting and labelling was the cause for confusion. Parent (all ages) meant any parent of children aged below 10, so a separate category to parents of adolescents aged 10-15. This has now been rectified in Table 1 to read “Parent (of child aged <10 years)”. The reason for the discrepancy was that some interviews included multiple interviewees, and all interviewees were only included when their data was used for analysis. We made an error in totalling up the count and a single participant (parent of a child aged <10) was not accounted for in the total participant count, but this has now been changed.

Our aim was to achieve breadth of experience by sampling a wide range of patients and professionals from various background and differing injuries. Unfortunately, we cannot report detail (e.g. age of professional participants) for such small numbers as this may risk identification of individuals. This work did not undertake sub-group analyses and felt that the detail reported is sufficient to balance confidentiality, participant burden and data available to us. As outlined in our recruitment strategy, we focussed on diversity in terms of age, gender, severity and cause of burn injury (for patient participants) and clinical experience or role (for professional participants) which we monitored and reported on.

Comment 3.8

Comparison of measures in burns trials

I have to confess to not understanding the purpose of this comparison. Why not compare to other conceptual work around outcomes and outcome domains in burns? The comparison provided seems to just give examples of measures of the ‘themes’ described. It doesn’t provide a complete review of how the concepts have been measured and using what tools and whether that is comprehensive or not. As such I am not sure of the ultimate utility of this comparison. Why is there not a comparison to existing studies that outline outcome priorities / domains in burns. A comparison of conceptualisation and contents and justification for this. How is this similar or different? How does it add to existing knowledge? I think the paper really needs to do this if it is proposing a categorisation of things that important to patients as a way of justifying this to reflect on outcome assessment going forward.

As mentioned earlier I do think that the unique element in this paper is potentially the HCP interviews – perhaps as a standalone piece of work that might add to the literature.

This study did not set out to complete a full review of concepts and their measurement instruments in burns research which could be considered a standalone piece of work which is in progress [11]. As mentioned in response to comment 3.1, this paper did not aim to propose a categorisation and extensive comparison with previous work is therefore outside the scope of this work. We have used existing literature to identify which outcomes overlap with themes identified in our research. In doing so, we highlighted 3 themes that we were unable to identify in the current literature (incl. existing outcome frameworks).

The purpose of mapping themes to outcomes was to investigate whether priorities align with routinely assessed outcomes, which other reviewers commented on, is a strength of our work (Comment 1.1 and 1.19). So it was necessary to contextualise our findings within the wider literature. We have added some detail to further explain the utility of the mapping on p. 8: “This step will contextualise identified themes within the current literature.”

Comment 3.9

Findings presented

These categories are very broad. Reading them and the contents I did think that this is only partially developed and again needs justification in relation to existing categorisations and frameworks. I was caused to question the thematic / category consistency. Two examples (1) Pain is reported as a most important priority but within this there is mention of itching, scar tightness, contractures and 'discomfort' – these are different to pain. (2) Psychological well-being – what is this and how is it defined and conceptualised? This is a really broad category and I'm afraid seemingly poorly described and conceptualised. Again this is difficult and complex but mentions PTSD, psychological well being, coping, worry, anxiety, guilt – again complex issues under a very simple category heading. Again I have to admit to a potentially skewed view here having spent much time myself analysing and organising conceptual categories in burns research. However, I am really concerned that a new categorisation, that doesn't refer or related to previous attempts, and is not that well justified, potentially clouds the water further.

We acknowledge and are aware of the work that has investigated many complex facets of burn recovery and outcome domains that include several nuanced concepts as demonstrated in previous outcome research in burns care and general trials [21,22].

This work represents a starting point and the broad categorisation is an aim of this work, and required due to our wide sampling strategy.

We could have adopted a more granular approach in our theme development to separate out e.g. itching, scar tightness, contractures, which were all coded separately during the analysis, but combined in iterative rounds of interpretation. A granular presentation would be unhelpful to demonstrate simply and clearly the variation of importance of outcome domains over time.

For example, our study found that, pain in general, is important to patients across short, medium and long term recovery. As the reviewer pointed out, we included sub-categories to be able to broadly demonstrate that the overarching outcome domain is represented in all phases of patients' recovery. Future work may wish to examine further how importance of such nuanced experiences within a specific outcome domain change over time. For example, contractures are likely to be relevant only during long-term recovery but as we mentioned in the discussion, it would be helpful for any future work which aims to explore time sensitivity of particular concepts, but any such research need to focus on more homogenous patient groups for which this concept is relevant. Similarly, pain is multi-faceted domain and sub-domains may include scar pain or donor site pain. However, further investigation warrants separate research studies which solely investigate variability of pain. We have now included in the discussion on p. 19 that the present work can be considered a starting point for research that might wish to further explore variability of granular concepts.

“This work provided a starting point to understanding the variability of the most important and broad priorities of patients and professionals during burn recovery. Additional qualitative work might assess more granular aspects of themes identified in this work amongst more homogenous patient groups. This can include, for instance, exploring more nuanced psychological concepts that are important to specific patient groups (e.g. adolescents or ethnic minority patients) or specific injury patterns (e.g. large burns) over time. Likewise, priorities important across all stages of recovery require further detailed examination to be able to inform specific outcome measurement. For example, different concepts related to pain (scar pain, burn wound pain, pain during the procedure) might be relevant at different time points.”

Again, we wish to point out that this work did not aim to produce a new categorisation or taxonomy.

Priorities – you say most important here. What does that mean and how was it determined? The list of ‘priorities’ seems very limited compared to the range of issues we know are important to patients and parents. Again at risk of self-citation I would refer to the comprehensive list of outcome domains and items referenced in my own work and qualitative systematic review (spanning 41 qualitative studies).

Thank you for highlighting this which was also raised in comment 1.4 and addressed in the relevant response. As mentioned in our response above, the main aim was not to elucidate the nuances of every concept and repeat the existing work that has been thoroughly synthesised already, but rather to demonstrate the magnitude of importance of some overarching themes at different time points as we have explained above.

Comment 3.10

Discussion

As mentioned above there is really no placement of this work within current knowledge and existing studies that present domain frameworks and categories.

Thank you but we hoped to have clarified in earlier responses to the reviewer’s comment that this is not relevant because we have not attempted to develop a new domain framework and categories. This study is aimed to be a starting point for discussions about the importance of timing of outcomes in burn recovery and not to extend, critique or endorse any domain frameworks and categories.

Comment 3.11

Limitations

Were there any limitations to conducting these in hospitals. Personally I have often found time limited and context associated with clinical care and sometimes therefore discussion is stilted. These are quite short interviews – between 15 and 45 minutes – what implications does that have e.g. depth / richness.

We thank the reviewer for sharing his own experiences with conducting interviews in hospitals. We have had no such impacts on discussions and can therefore not report this as a limitation. We included in our data collection section on p. 5 that we selected a “quiet space” in hospitals and have now expanded this sentence to explicitly state that we prevented any impact by design: “Face-to-face interviews were carried out in a quiet and private space in one of the four hospitals to ensure no interruptions or time limitations were impacting the interviews.”

The range given for length of interview reports the minimum and maximum for both participant groups, so reports only outliers. We have now included the average (mean) of 32 minutes to show that most interviews lasted longer. We also reported pooled results here and it should be noted that interviews with professionals were shorter (mean= 29 min) to reduce burden.

We would also like to point out that the length of the interview highly depends on the individual. Some participants prefer longer narratives of their experience and for others, it is difficult to elaborately express their views. This does, however, not necessarily impact the quality (or richness) of data. As outlined in our description of saturation, interviews were stopped when data collection satisfactorily

answered our research question, to which all data collection contributed, irrespective of their length. We acknowledge that some qualitative work reports longer interviews (e.g. 60min, [29]; 42-85min [30]) whereas other studies report similar length to ours (e.g. 30min, [31]). We were interested in the concepts most important to patients and professionals across time, so were interested in breadth. Whilst prompts were used to elicit richness of priorities mentioned, less depth was required for those concepts not considered a priority.

Comment 3.12

There doesn't seem to be any reflection on imitations of a non-longitudinal approach (sequential over time) or lack of specific sampling based on time point since injury / acute care?

Thank you, and we acknowledge that further detail was needed to explain that we did not employ a longitudinal approach in this study. Data collection was conducted at a single time point and participants were asked to reflect upon their experience at a single time point. As highlighted in response to comments 3.2 and 3.4, we have clarified this in response to another reviewer's comment (1.4) and have also addressed this in the limitations section on p.18 by including "time since injury" as an additional confounding factor amongst other contextual complexities highlighted.

Comment 3.13

I am sorry if this review comes over as overly negative. I do think the novel angle here is the HCP interviews given time and effort to develop their perspectives on how working within and between services and supporting patients can affect outcomes. That seems novel considering existing knowledge.

We thank the reviewer for the helpful suggestions which led to improvements in the manuscript. We also appreciated the reviewer considering the findings from our interviews with professionals as valuable addition to the current body of knowledge. We hope to have clarified the misunderstanding that prompted several comments: this work is not attempting to replicate or overthrow any existing outcome domain frameworks and categories, but we wish to highlight that priorities differ over time and differ between burn survivors/patients and professionals.

References

1. Sebele-Mpofu FY. Saturation controversy in qualitative research: Complexities and underlying assumptions. A literature review. <http://www.editorialmanager.com/cogentsocsci> [Internet] Cogent; 2020 Jan 1 [cited 2022 May 12];6(1). [doi: 10.1080/23311886.2020.1838706]
2. Fusch PI, Ness LR. Are We There Yet? Data Saturation in Qualitative Research. Qual Rep [Internet] 2015 [cited 2022 May 12];20(9):1408–1416. Available from: <http://www.nova.edu/ssss/QR/QR20/9/fusch1.pdf>
3. Hennink MM, Kaiser BN, Marconi VC. Code Saturation Versus Meaning Saturation: How Many Interviews Are Enough? Qual Health Res [Internet] SAGE Publications Inc.; 2017 Mar 1 [cited 2022 May 12];27(4):591–608. PMID:27670770
4. Hennink M, Kaiser BN. Sample sizes for saturation in qualitative research: A systematic review of empirical tests. Soc Sci Med Pergamon; 2022 Jan 1;292:114523. PMID:34785096

5. Braun V, Clarke V. Using thematic analysis in psychology. *Qual Res Psychol* [Internet] 2006 [cited 2020 Jun 29];3(2):77–101. [doi: 10.1191/1478088706qp063oa]
6. Pham TN, Kramer CB, Wang J, Rivara FP, Heimbach DM, Gibran NS, Klein MB. Epidemiology and outcomes of older adults with burn injury: An analysis of the national burn repository. *J Burn Care Res* [Internet] *J Burn Care Res*; 2009 Jan [cited 2021 Jun 20];30(1):30–36. PMID:19060727
7. Williams NR, Davey M, Klock-Powell K. *Rising from the Ashes*. http://dx.doi.org/101300/J010v36n04_04 [Internet] Taylor & Francis Group ; 2008 [cited 2022 May 23];36(4):53–77. PMID:12836780
8. Jones BA, Buchanan H, Harcourt D. The experiences of older adults living with an appearance altering burn injury: An exploratory qualitative study. *J Health Psychol* [Internet] SAGE Publications Ltd; 2017 Mar 1 [cited 2021 Jun 20];22(3):364–374. PMID:26324235
9. Spies M, Herndon DN, Rosenblatt JI, Sanford AP, Wolf SE. Prediction of mortality from catastrophic burns in children. *Lancet Elsevier B.V.*; 2003 Mar 22;361(9362):989–994. PMID:12660055
10. Robert R, Meyer W, Bishop S, Rosenberg L, Murphy L, Blakeney P. Disfiguring burn scars and adolescent self-esteem. *Burns Elsevier*; 1999 Nov 1;25(7):581–585. PMID:10563682
11. Young A, Brookes S, Rumsey N, Blazeby J. Agreement on what to measure in randomised controlled trials in burn care: Study protocol for the development of a core outcome set. *BMJ Open* BMJ Publishing Group; 2017 Jun 1;7(6):e017267. [doi: 10.1136/bmjopen-2017-017267]
12. Falder S, Browne A, Edgar D, Staples E, Fong J, Rea S, Wood F. Core outcomes for adult burn survivors: A clinical overview. *Burns Elsevier*; 2009. p. 618–641. PMID:19111399
13. Hughes KL, Clarke M, Williamson PR. A systematic review finds Core Outcome Set uptake varies widely across different areas of health. *J Clin Epidemiol Pergamon*; 2021 Jan 1;129:114–123. PMID:32987162
14. Mathers J, Moiemmen N, Bamford A, Gardiner F, Tarver J. Ensuring that the outcome domains proposed for use in burns research are relevant to adult burn patients: A systematic review of qualitative research evidence. *Burn Trauma* [Internet] 2021 [cited 2021 Jun 21];8:30. [doi: 10.1093/BURNST/TKAA030]
15. Simons M, Lim PCC, Kimble RM, Tyack Z. Towards a clinical and empirical definition of burn scarring: A template analysis using qualitative data. *Burns Elsevier*; 2018 Nov 1;44(7):1811–1819. PMID:30060903
16. Williamson PR, Altman DG, Blazeby JM, Clarke M, Devane D, Gargon E, Tugwell P. Developing core outcome sets for clinical trials: Issues to consider. *Trials* [Internet] BioMed Central; 2012 Aug 6 [cited 2021 Jun 21];13(1):1–8. PMID:22867278
17. Nice. children and young people’s healthcare [Internet]. 2020. Available from: <https://www.nice.org.uk/Media/Default/About/what-we-do/Into-practice/measuring-uptake/children-young-people-impact-report/nice-impact-children-young-people-healthcare.pdf>
18. Stylianou N, Buchan I, Dunn KW. A review of the international Burn Injury Database (iBID) for England and Wales: descriptive analysis of burn injuries 2003-2011 A review of the international Burn Injury Database (iBID) for England and Wales: descriptive analysis of burn injuries. *BMJ Open* [Internet] 2003 [cited 2022 May 17];5:6184. [doi: 10.1136/bmjopen-2014]
19. Davies K, Johnson EL, Hollén L, Jones HM, Lyttle MD, Maguire S, Kemp AM. Incidence of medically attended paediatric burns across the UK. *Inj Prev* [Internet] BMJ Publishing Group; 2020 Feb 1 [cited 2022 May 17];26(1):24. PMID:30792345
20. Braun V, Clarke V. *Successful qualitative research : a practical guide for beginners*. SAGE; 2013. ISBN:1847875823

21. Dodd S, Clarke M, Becker L, Mavergames C, Fish R, Williamson PR. A taxonomy has been developed for outcomes in medical research to help improve knowledge discovery. *J Clin Epidemiol* [Internet] Elsevier USA; 2018 Apr 1 [cited 2020 Jul 23];96:84–92. PMID:29288712
22. Young AE, Davies A, Bland S, Brookes S, Blazeby JM. Systematic review of clinical outcome reporting in randomised controlled trials of burn care. *BMJ Open* [Internet] British Medical Journal Publishing Group; 2019 Feb 1 [cited 2021 Nov 16];9(2):e025135. PMID:30772859
23. Young AE, Brookes ST, Avery KNL, Davies A, Metcalfe C, Blazeby JM. A systematic review of core outcome set development studies demonstrates difficulties in defining unique outcomes. *J Clin Epidemiol* Pergamon; 2019 Nov 1;115:14–24. PMID:31276780
24. Jones LL, Calvert M, Moiemmen N, Deeks JJ, Bishop J, Kinghorn P, Mathers J. Outcomes important to burns patients during scar management and how they compare to the concepts captured in burn-specific patient reported outcome measures. *Burns* Elsevier Ltd; 2017 Dec 1;43(8):1682–1692. PMID:29031889
25. Bunniss S, Kelly DR. Research paradigms in medical education research. *Med Educ* [Internet] John Wiley & Sons, Ltd; 2010 Apr 1 [cited 2021 Aug 2];44(4):358–366. [doi: 10.1111/J.1365-2923.2009.03611.X]
26. Patton MQ. *Qualitative research & evaluation methods : integrating theory and practice*. SAGE Publications; 2002. ISBN:9781412972123
27. Chen X, Zhang T, Wang H, Feng Z, Jin G, Shao S, Du J, Cn CE. Factors influencing the prescription pattern of essential medicines from the perspectives of general practitioners and patients: a qualitative study in China. *BMJ Open* [Internet] 2022 [cited 2022 May 19];12:55091. [doi: 10.1136/bmjopen-2021-055091]
28. Curry L, Ayedun A, Cherlin E, Taylor B, Castle-Clarke S, Linnander E. The role of leadership in times of systems disruption: a qualitative study of health and social care integration. *BMJ Open* [Internet] British Medical Journal Publishing Group; 2022 May 1 [cited 2022 May 19];12(5):e054847. PMID:35568492
29. Li Y, Zhou L, Tang L, Liu M, Ming X, Shen F, Cui J, Meng X, Zhao J. Burn patients' experience of pain management: A qualitative study. *Burns* Elsevier; 2012 Mar 1;38(2):180–186. PMID:22079543
30. Mirlashari J, Nasrabadi AN, Amin PM. Living with burn scars caused by self-immolation among women in Iraqi Kurdistan: A qualitative study. *Burns* Elsevier; 2017 Mar 1;43(2):417–423. PMID:28341263
31. Rencken CA, Harrison AD, Aluisio AR, Allorto N, Harrison AA; D;, Aluisio AR;, Allorto NA. A Qualitative Analysis of Burn Injury Patient and Caregiver Experiences in Kwazulu-Natal, South Africa: Enduring the Transition to a Post-Burn Life. *Eur Burn J* 2021, Vol 2, Pages 75-87 [Internet] Multidisciplinary Digital Publishing Institute; 2021 Jul 1 [cited 2022 May 25];2(3):75–87. [doi: 10.3390/EBJ2030007]

VERSION 2 – REVIEW

REVIEWER	Mathers, Jonathan University of Birmingham College of Medical and Dental Sciences, Institute of Applied Health Research
REVIEW RETURNED	06-Jul-2022
GENERAL COMMENTS	I read the authors' revision and response to review comments with great interest. I do not feel that this is the substantive revision that was required to make this a robust and novel contribution to

	knowledge in this area. I refer to some authors' responses relating to my comments below. Response to comment 3.1. The authors state that they did not set out to comprehensively explore concepts and nuances within suggested themes / outcomes and that they have acknowledged this in the manuscript. By defining themes that relate to priorities (outcomes?) they are implicitly exploring outcome-related concepts which are important to patients and professionals. Further, by providing a comparison to examples of measures used in burns research they are expressly comparing conceptual concept of those measures with the findings presented (as per the heading of the relevant table). Overall therefore, I do not agree with their assertion. Response to comment 3.2. Both I and another reviewer asked for clarification about the a priori time point codes. The response seems to imply that anything beyond 30 days post injury was considered as long term, but this is still not defined in the paper clearly. This is an overt aim of the research but no data relating to how long post injury patients were at the time of interview seems to have been collected. This makes the time point definitions somewhat vague and classifying 30 days post injury as long term seems questionable. The manuscript seems to be a little contradictory on this stating that all participants were 30 days post injury and could be considered in long term rehabilitation and then in data collection saying that medium term was 'around' the time of discharge and long term during rehabilitation (when?). As a reader of the paper I don't know how to interpret what is meant by short, medium and long term. I don't consider 30 days post injury to be long term and would expect that patients certainly do not. Comment 3.4 The response seems to claim that any a priori definition of purposive sampling criteria is inappropriate. I was simply asking for more information about what you were aiming for with your sample and particularly in relation to what seems to be the main aim of your research, to look at priorities over time. If you set out a priori to research that, you need to consider the appropriateness of your sample to look at differences in priorities over time i.e. does your sample have the relevant length of experience and can you demonstrate that to readers of your research. The fact that you cannot describe how long since injury your patient sample had experience of has to at least be acknowledged as a limitation unless you are claiming that anyone post day 30 regarding injury has long term experience of living with a burn injury, which as stated above I do not see the sense of. Comment 3.5. There are still no references or detail relating to the approach for saturation given in the revision and I can't see the additional detail on approach to saturation that the authors mention as being added to the revision. Comment 3.6. As noted already the manuscript still does not give any detailed definition of the short, medium and long term time points referred to. In terms of interpretation this is a severe limitation related to the
--	---

	aims and as noted regarding the sample makes it impossible to judge how appropriate the sample is to address those aims. Comment 3.7. Thankyou for the clarification regarding the sample characteristics. However, as noted having no information about how long since injury patient participants were, is a clear limitation to claims that the sample has experience of the relevant (undefined) medium and long term. Comments 3.8 and 3.9 I do not think the authors have adequately responded to my comments here. The very reason the analysis and conceptualisations, and robustness of these are so important is that the manuscript goes on to make comparisons with measures used in burns research (only via example, not comprehensively) and so readers potentially draw incorrect conclusions from this work. The authors state that they have combined codes into broad categories and reported these. The category headings, consistency and contents are absolutely crucial though, especially if reflecting on measures in burns care and the current coverage of things that are priorities to patients and professionals. I used the example of pain which contains things which are clearly not conceptually the same as pain, or at least would need to be carefully and clearly linked conceptually to pain. The authors have not addressed this at all in the revision. The table then gives an example of a measure of pain, but not the other within-category concepts described. I also identified psychological well-being as needing further definition and consideration. In the table the authors give an example of HADS which is specifically a within hospital assessment of anxiety and depression. Psychological well-being is far more broad than this (as described in other work and here) and many elements of 'psychological well-being' are not well considered in many existing burns measures. Yet the comparison given here implies that psychological well-being is considered in burns research as the HADS is an example of this. Highlighting HADS does not even cohere with the description given by the authors for this category which is largely about broader psychological and psychosocial impacts. As a result I don't see the utility of the comparisons given in this table and think there is significant potential for readers to draw incorrect conclusions about the use of measures in current burns research.
--	---

VERSION 2 – AUTHOR RESPONSE

Reviewer: 3 Dr. Jonathan Mathers, University of Birmingham College of Medical and Dental Sciences Comments to the Author: I read the authors' revision and response to review comments with great interest. I do not feel that this is the substantive revision that was required to make this a robust and novel contribution to knowledge in this area. I refer to some authors' responses relating to my comments below.
--

We thank the reviewer for their continued suggestions to improve the manuscript. These discussions are very helpful, and we fully welcome the opportunity to satisfy the reviewer's remaining issues.

Response to comment 3.1.

The authors state that they did not set out to comprehensively explore concepts and nuances within suggested themes / outcomes and that they have acknowledged this in the manuscript. By defining themes that relate to priorities (outcomes?) they are implicitly exploring outcome-related concepts which are important to patients and professionals. Further, by providing a comparison to examples of measures used in burns research they are expressly comparing conceptual concept of those measures with the findings presented (as per the heading of the relevant table). Overall therefore, I do not agree with their assertion.

Thank you. We understand the need to maintain the integrity of previous work that has uncovered outcome-related concepts important to patients and professionals and are highly respectful of the good work to improve and standardise outcome domains and measurement in this area.

We wish to avoid any confusion and have therefore made significant revisions to our submission in response to the present comment and to the final comment ('Comments 3.8 and 3.9'). We have revisited the raw data and thoroughly reviewed our approaches to mapping themes to outcomes in team discussions. The following major changes were made:

We have now supplied an additional supplementary file with full descriptive reports that were produced as an interim step during analyses to provide detail about the themes and sub-themes identified from the analysis of interviews with patients and professionals. They provide additional detail for interested readers who wish to examine and understand the full content of particular themes. In addition, we have also included two tables (Table S4 and S5) in the same supplementary file that outline more detail about how original themes are linked to combined themes. We hope that by including descriptive reports and detailed accounts of analysis steps, we provide full transparency to those wishing to cross-reference the reported summarised results.

The study produced a wealth of data from interviews with 52 participants. Two separate analyses for two stakeholder groups (patients and professionals) were conducted and an additional layer of analyses and refinement was needed to combine themes. Part of this process is condensing information and relabelling themes to produce an informative account of the qualitative experiential data. We therefore agree that some theme labels may appear broad. When reviewing of the raw data, we revisited the two themes identified by the reviewer as needing attention ('pain' and 'psychological wellbeing'). Upon review, we decided to change this to 'pain and discomfort' to reflect experiences of tightness and itching which participants described as painful and uncomfortable. Changes were implemented throughout the manuscript, Figures and tables. We felt, however, that no changes to the theme 'psychological wellbeing' were appropriate as it accurately captures the different codes identified and language used by participants. Again, we feel that by providing the full descriptive report, interested readers can cross-reference the content of the theme if they wish.

2) We have amended our approach to mapping themes to outcomes. Instead of mapping outcomes by using an exemplary approach, we have used an existing outcome classification for burn care research developed through synthesis of 955 unique outcomes reported across 147 trials in burn care [1] (further detail in the response to 'Comments 3.8 and 3.9'). Themes and their content identified in both groups were each mapped to outcome domains listed in the classification system and reported in detail in tables S4 and S5. Table 2 in the results section (p. 17) now shows the summary of outcome domains mapped to combined themes. The outcome classification system is also included in a separate supplementary file for ease of access (Table S3).

We have made revisions to the whole methods section to reflect these changes, clarifying methods and wording, and signposting to the supplementary files. The section on pp.7-8 now reads:

"Analysis of qualitative data was undertaken using a qualitative data management software (NVivo Version 12). In a first step, transcripts were analysed using thematic analysis [2] to understand

priorities of patients and professionals (study objective i). Interview transcripts were analysed separately for patients and professionals to accommodate concepts unique to each group and facilitate later examination of differences. This study used deductive and inductive approaches to coding [3]. Codes for short, medium and long term priorities were developed *a priori*, to explore perspectives over time (study objective ii). *A priori* codes corresponded to time points (short term, medium term, long term) defined for the purpose of this study (described above). Additional codes were identified through line-by-line coding following principles of inductive theme development in line with six steps of thematic analysis [2]. Transcripts were read and re-read to allow familiarisation with the data. Initial codes were assigned from which themes were developed: Coded excerpts were grouped into themes of similar meaning and organised into a hierarchical structure of themes and sub-themes. This thematic structure was refined as new themes emerged. These were then reviewed and refined through discussions amongst the study team until final definitions and labels were agreed: This process involved iterative rounds of interpretation of contexts specific to the interviewees' injury and identification of relationships between themes. Diagrams illustrating themes and relationships were used to visualise the data and aid discussions between team members. In a second step, differences in priorities between patients and professionals were examined by comparing narrative accounts of each group. Connections were drawn between patients and professionals by qualitatively contrasting themes identified for each group to find similarities and differences in their perspectives (study objective i). This step also involved combining themes by condensing and relabelling which was shaped by the data and the team's experience and knowledge with defining, identifying and classifying outcomes in burns research.

Coding and analyses were undertaken independently by two researchers (CH, PD) who met regularly to discuss emerging themes from the data. Close contact between the coders during data analyses ensured consistency in coding and agreement of interpretation of the data. Independent double-coding was also performed for 25% of transcripts by the two reviewing authors. A purposive sample of transcripts that underwent double-coding was used to ensure maximum variation by participant group, socio-demographic background and injury severity. Data from double-coding were compared and discussed to ensure consistency in approaches to further coding. Any uncertainties and discrepancies during this process were discussed with the wider study team and in consultation with an experienced qualitative researcher who was independent to the study team (DE). Interim interpretation and structure of themes was discussed with the wider study group and if necessary, iteratively modified.

Analyses were carried out in parallel to data collection to determine data saturation separately for both participant groups. This means, no new codes or meaning were identified through additional interviews and sufficient data was collected to address the research objectives [4–7]. Specifically, interim results during analyses were reviewed regularly by the study team and decisions of whether saturation was achieved were based on impressions of whether collected data provides sufficient conceptual depth [8,9]. If saturation was expected, a small number of additional interviews were conducted to confirm that no new codes or meaning were identified [10]. This process was overseen by the senior author (AY) who has a clinical background in burn care. A final thematic structure and

meaning of themes were agreed in team discussions. Findings were collected in two descriptive accounts (one for patients, one for professionals) to be able to identify similarities and differences in key findings.

Themes identified through analyses described above were mapped against common outcomes in trials to assess whether perceptions explored in this work align with routinely measured outcomes in burns research (objective iii). This step involved comparing identified themes and their conceptual content to an existing outcome classification (see supplementary file, Table S3) which was developed through synthesis of 955 unique outcomes reported across 147 randomised controlled trials in burn care [1].

This multi-layer analysis allowed to elicit priorities qualitatively from experiences of patients and professionals while linking experiential data to previously identified and classified outcomes.”

It is worth noting that we agree that through exploring participants' priorities (outcomes), that there will be some overlap with existing work that has comprehensively identified concepts relating to a single outcome domain. And we do not wish to suggest that this isn't the case. However, we do maintain that we did not aim to identify outcome-related concepts comprehensively and exhaustively. Our research objectives did not claim to contradict or augment existing work to refine understanding of outcome (measurement) by eliciting further concepts relevant to particular domains. Overarching themes identified here were inductively identified from the data collected for the purposes of this study. It is therefore possible that conceptual content of themes may vary to results of inductive thematic analysis reported in other work. For example, some work has dedicated qualitative inquiry to concepts related to experiences with scarring [11–13] or pain management [14]. We would consider this work to comprehensively explore nuances and outcome-related concepts and enormously helpful to inform sophisticated measurement of outcome domains. The focus of this work, however, was to provide a starting point to investigate time sensitivity of outcome domains. We set out to investigate the breadth of priorities rather than focus on the depth of a single outcome domain. The themes identified here may well represent concepts explored in other work, but the activities described in the current work were never able to comprehensively identify all nuanced concept in the time of the interviews conducted in the present work. We appreciate this is a worthwhile endeavour and we hope that this work will offer a springboard for future work to invest resources to explore time sensitivity of concepts within outcome domains.

We also content that the study provides an important contribution to knowledge in that the current study i) contrasted patients' and professionals' views, identifying differences and similarities in their priorities, ii) highlighted at which time points these priorities are salient for both groups and iii) identified potentially important priorities currently not considered in commonly measured outcomes in burn care research. We were unable to identify research that provides any such insights.

Response to comment 3.2.

Both I and another reviewer asked for clarification about the a priori time point codes. The response seems to imply that anything beyond 30 days post injury was considered as long term, but this is still not defined in the paper clearly. This is an overt aim of the research but no data relating to how long post injury patients were at the time of interview seems to have been collected. This makes the time point definitions somewhat vague and classifying 30 days post injury as long term seems questionable. The manuscript seems to be a little contradictory on this stating that all participants were 30 days post injury and could be considered in long term rehabilitation and then in data collection saying that medium term was 'around' the time of discharge and long term during rehabilitation (when?). As a reader of the paper I don't know how to interpret what is meant by short, medium and long term. I don't consider 30 days post injury to be long term and would expect that patients certainly do not.

Comment 3.6.

As noted already the manuscript still does not give any detailed definition of the short, medium and long term time points referred to. In terms of interpretation this is a severe limitation related to the aims and as noted regarding the sample makes it impossible to judge how appropriate the sample is to address those aims

We thank the reviewer for raising these points and we acknowledge that changes in wording following our last iteration of the manuscript resulted in confusion and contradiction. We fully agree with the reviewer that 30 days post injury does not indicate long term recovery, but in the absence of reliable data on injury date, we can confidently report that all patients were at least 30 days post injury and/or not in acute care. We have removed any reference to indicate that 30 days would be defined as “long term”. We have added clarification in the methods section which now reads:

“Patients with varying injury severity and burn sizes were eligible. Interviews were conducted at least 30 days after injury and/or with patients who were not in acute treatment (e.g. larger burns). This ensured that all participants were able to reflect on their experiences around the time when the injury happened and during acute treatment, and that they were also able to articulate what is most important to them when in the following stages of recovery.”

There is no single definition of short, medium or long term in the literature and empirical work is needed to define meaningful time points for a classification of short, medium and long term. Different uses of the terms short and long term exist, which may also vary for different outcomes [15]. For instance, studies investigating impacts of electrical burn have defined long-term as 6-24 months post injury [16] and others have defined long-term as 6 years post injury [17]. Given the breadth of perspectives and variety of injuries investigated, we felt it is unhelpful to a single definition that applies to all participants. In the absence of a common definition, we defined timeframes in terms of significant stages during recovery. Point of discharge was referenced as a crucial point for outcome assessment [15], so we have decided to use this as middle points.

We have further elaborated on this in the methods on p. 6 by stating “Participants were asked to reflect on their experience or priorities relating to different times after burn injury. There are no consensus definitions of time frames in burn care. To ensure consistency of explored perspectives, significant points during recovery from a burn injury were used as temporal anchors (e.g. point of discharge is considered a crucial point for outcome assessment [15]). For the purpose of this study, time periods were therefore defined *a priori* as: short term (immediately after the injury), medium term (from around the time of discharge and/or wound healing) and long term (during rehabilitation).”

Further significant additions and changes to the relevant discussion section were made on pp. 18-19 which now reads:

“Other limitations should be acknowledged. First, our sample consisted of patients with a variety of burn size which has implications for elicited perceptions across different time frames investigated. For instance, patients with larger burns experience longer healing [18]. This study defined time frames (short, medium and long term) *a priori* which were agreed after expert consultation and discussions within the multi-disciplinary study team. Individuals’ definitions of short, medium and long term recovery, however, are highly personal and might depend on contextual complexities (e.g. size and location of burn, time since injury, socio-demographic details of patients, any wound complications), and might therefore differ to the definitions used in this study. Challenges to standardise time frames for recovery from burn injury are recognised considering the variabilities and complexities of injuries [18–20]. Long term outcome measurement, for example, can range from a few months [21] to years [22] across studies assessing functional outcomes for large burns. Further research is warranted to pursue consensus definitions of time frames in burn recovery. In addition, time periods between participants’ time of injury and interview varied across the sample, but no systematic variation in length of experience of recovery was pursued during recruitment. Due to resource limitations and practical considerations, the current work also did not employ a longitudinal study design but merely provides a snapshot of participants’ perceptions. Quantitative and qualitative evidence suggests that varying recall periods may influence health services research [23,24]. We were unable to collect reliable data on injury date to be able to assess the impact of potential recall bias caused by varying intervals on the results.”

Comment 3.4

The response seems to claim that any a priori definition of purposive sampling criteria is inappropriate. I was simply asking for more information about what you were aiming for with your sample and particularly in relation to what seems to be the main aim of your research, to look at priorities over time. If you set out a priori to research that, you need to consider the appropriateness of your sample to look at differences in priorities over time i.e. does your sample have the relevant length of experience and can you demonstrate that to readers of your research. The fact that you cannot describe how long since injury your patient sample had experience of has to at least be acknowledged as a limitation unless you are claiming that anyone post day 30 regarding injury has long term experience of living with a burn injury, which as stated above I do not see the sense of.

Comment 3.7.

Thankyou for the clarification regarding the sample characteristics. However, as noted having no information about how long since injury patient participants were, is a clear limitation to claims that the sample has experience of the relevant (undefined) medium and long term.

We also agree with the reviewer that it would be informative to report time since injury. We attempted to collect this data, but faced issues leading to a large amount of incomplete data. In particular, we were able to obtain only 13 out of 32 data points for “date of injury” (dates available to us ranged from 1992 – September 2019; so a considerable variation in time). We therefore considered reporting this information unhelpful, but are happy to take editorial guidance on this.

It is worth noting that, in our previous response, we meant that specific configuration and characteristics of the sample would be impractical to define a priori. We aimed to achieve variation between participants in terms of age, gender, severity, and cause of burn and have reported all data that were available to us. Indeed, as Table 1 shows, the sample contained patients with 7 different causes of burned of minor, moderate or major severity. This variation also implies differences in acute treatment length which means that recruitment only occurred after a considerable length of time.

We also did not include “time since injury” as a sampling criterion because we felt it would be unrealistic to adhere to defined criteria. Instead, we felt it is reasonable to assume that variation would be achieved naturally during recruitment. We have acknowledged this in the discussion section along with our changes made in response to the previous comments 3.4 and 3.7.

Comment 3.5.

There are still no references or detail relating to the approach for saturation given in the revision and I can't see the additional detail on approach to saturation that the authors mention as being added to the revision.

Thank you for pointing this out. In our response to 3.4 we have referred to an earlier comment by reviewer 1 (comment 1.8 in revision 1) in which we have detailed our approach to saturation and included additional references. Indeed, our response stated that

“No formal analysis for saturation was undertaken as we agree with current opinions about unsuitability of determining saturation a priori [25]. In absence of formal analyses, it is not suitable to cite a specific method to determine saturation but explained which approach we followed. We therefore further argued that: “Instead, we chose to adopt approaches that focus on the saturation of both, codes and meaning and how sufficiently the data addresses the research objectives [4,5].” In our previous response we also referred to changes made to the analysis section where we cited above references: “Analyses were carried out in parallel to data collection to determine data saturation. This means, no new codes or meaning were identified in the interviews and sufficient data was collected to address the research objectives [4,5].” We have now added two additional references to support this approach [6,7].

Our approach is in line with authors who have critically examined the concept of saturation and discourses around “sufficient conceptual depth”[8,9] or “theoretical sufficiency” [26]. We have added further detail to further explain this process on p. 8. This includes the fact that saturation was determined separately for both groups, explicit clarification that data review sought conceptual depth and that further sampling was undertaken if data saturation was expected and have added references where similar approaches were reported [10]. The section now reads:

“Analyses were carried out in parallel to data collection to determine data saturation separately for both participant groups. This means, no new codes or meaning were identified through additional interviews and sufficient data was collected to address the research objectives [4–7]. Specifically, interim results during analyses were reviewed regularly by the study team and decisions of whether saturation was achieved were based on impressions of whether collected data provides sufficient conceptual depth [8,9]. If saturation was expected, a small number of additional interviews were conducted to confirm that no new codes or meaning were identified [10].”

We have also added to the “setting and participant” section on p.6 to signpost to the relevant section where this is described in more detail.

In our previous response, we also highlighted that “a recent systematic review concluded that 9-17 interviews are necessary to reach saturation [27].” An additional review of qualitative studies in burn care found an average of 22 participants per study [28]. We therefore remain confident that sufficient data was collected given that our sample aligns with previous work (32 patient participants, 21 professional participants).

Comments 3.8 and 3.9

I do not think the authors have adequately responded to my comments here. The very reason the analysis and conceptualisations, and robustness of these are so important is that the manuscript goes on to make comparisons with measures used in burns research (only via example, not comprehensively) and so readers potentially draw incorrect conclusions from this work. The authors state that they have combined codes into broad categories and reported these. The category headings, consistency and contents are absolutely crucial though, especially if reflecting on measures in burns care and the current coverage of things that are priorities to patients and professionals. I used the example of pain which contains things which are clearly not conceptually the same as pain, or at least would need to be carefully and clearly linked conceptually to pain. The authors have not addressed this at all in the revision. The table then gives an example of a measure of pain, but not the other within-category concepts described. I also identified psychological well-being as needing further definition and consideration. In the table the authors give an example of HADS which is specifically a within hospital assessment of anxiety and depression. Psychological well-being is far more broad than this (as described in other work and here) and many elements of ‘psychological well-being’ are not well considered in many existing burns measures. Yet the comparison given here implies that psychological well-being is considered in burns research as the HADS is an example of this. Highlighting HADS does not even cohere with the description given by the authors for this category which is largely about broader psychological and psychosocial impacts. As a result I don’t see the utility of the comparisons given in this table and think there is significant potential for readers to draw incorrect conclusions about the use of measures in current burns research.

We thank the reviewer for explaining the issue and want to avoid readers drawing incorrect conclusions. We have addressed this comment in our first response (Response to comment 3.1) and have significantly revised steps to map themes to outcomes (objective iii).

We feel that this step remains important to link themes to outcomes which was endorsed by both reviewers in the previous review. We have outlined throughout our manuscript, and added descriptions, to explain in more detail the rationale and utility. Specifically, in the introduction (p.4): “It is therefore important to understand professionals’ and patients’ priorities during burn recovery as both need to play a role in making decisions about treatment and care [29,30]. Likewise, understanding how they are linked to commonly measured outcomes in burn care research may

guide future study design.” In the methods (p. 8) “This multi-layer analysis allowed to elicit priorities qualitatively from experiences of patients and professionals while linking experiential data to previously identified and classified outcomes.” And the discussion (p. 20): “An aim of on-going research is to ensure that frequently measured outcomes adequately capture all important priorities related to burns recovery [31]. The results from mapping themes to commonly measured outcomes in burn care research highlighted potentially important priorities currently not considered. . Specifically, perceptions related to ‘uncertainty’ (patients), ‘sense of control’ and ‘patient knowledge, understanding & support’ (professionals) were all identified as important priorities at varying time points during recovery. The existence and impacts of uncertainty on patients are well known [32] and, as supported by our findings, and have also been linked to the level of information provision [33,34]. Conversely, professionals in our study highlighted that ‘patient knowledge, understanding & support’ is one of their key priorities during the medium term of patients’ recovery from a burn injury. Routine assessment of perceptions relevant to level of uncertainty and information provision as well as assessing patients’ level of knowledge and understanding and support received can be important beyond commonly measured outcomes.”

We used the previous approach of mapping themes to outcomes to illustrate comparisons by example rather than comprehensively, because we did not feel any existing taxonomies were accounting for some of the nuances in the data and because it was beyond the scope of this work to complete a systematic review of outcomes and their measurement instruments in burns care. We have now used the, to our knowledge, most comprehensive outcome classification available, which was developed through a systematic review of outcomes reported in randomised controlled trials in burn. We felt this was most informative as using a standardised classification system will facilitate ease of comparison, interpretation, and replication.

The challenge of adequately measuring concepts using available measurement instruments has been recognised and we are highly respectful of the reviewer’s own work to drive forward research to homogenise this important area of work. By comparison current practice of reporting, we are not recommending that themes should be measured in a certain way and have made this explicit in the discussion on p. 22 “Indeed, this work does not attempt to recommend how outcomes should be measured and further in-depth work is required for this.”

We hope that the substantial changes throughout the manuscript satisfy the reviewer’s remaining issues. Further small changes to the manuscript were made to improve clarity, correct typos and add acknowledgements.

References

1. Young AE, Davies A, Bland S, Brookes S, Blazeby JM. Systematic review of clinical outcome reporting in randomised controlled trials of burn care. *BMJ Open* [Internet] British Medical Journal Publishing Group; 2019 Feb 1 [cited 2021 Nov 16];9(2):e025135. PMID:30772859
2. Braun V, Clarke V. Using thematic analysis in psychology. *Qual Res Psychol* [Internet] 2006 [cited 2020 Jun 29];3(2):77–101. [doi: 10.1191/1478088706qp063oa]
3. Joffe H, Yardley L. Content and thematic analysis. In: Marks D, Yardley L, editors. *Res methods Clin Heal Psychol* London: Sage; 2004. p. 56–69.
4. Fusch PI, Ness LR. Are We There Yet? Data Saturation in Qualitative Research. *Qual Rep* [Internet] 2015 [cited 2022 May 12];20(9):1408–1416. Available from: <http://www.nova.edu/ssss/QR/QR20/9/fusch1.pdf>
5. Hennink MM, Kaiser BN, Marconi VC. Code Saturation Versus Meaning Saturation: How Many Interviews Are Enough? *Qual Health Res* [Internet] SAGE Publications Inc.; 2017 Mar 1 [cited

- 2022 May 12];27(4):591–608. PMID:27670770
6. Saunders B, Sim J, Kingstone T, Baker S, Waterfield J, Bartlam B, Burroughs H, Jinks C. Saturation in qualitative research: exploring its conceptualization and operationalization. *Qual Quant* [Internet] Springer; 2018 Jul 1 [cited 2022 Aug 25];52(4):1893. PMID:29937585
 7. Fofana F, Bazeley P, Regnault A. Applying a mixed methods design to test saturation for qualitative data in health outcomes research. *PLoS One* [Internet] Public Library of Science; 2020 Jun 1 [cited 2022 Sep 30];15(6). PMID:32559246
 8. Nelson J. Using conceptual depth criteria: addressing the challenge of reaching saturation in qualitative research: <https://doi.org/10.1177/1468794116679873> [Internet] SAGE PublicationsSage UK: London, England; 2016 Dec 14 [cited 2022 Aug 25];17(5):554–570. [doi: 10.1177/1468794116679873]
 9. Morse JM. Data were saturated... *Qual Health Res* [Internet] SAGE Publications Inc.; 2015 May 4 [cited 2022 Aug 25];25(5):587–588. PMID:25829508
 10. Morse WC, Lowery DR, Steury T. Exploring Saturation of Themes and Spatial Locations in Qualitative Public Participation Geographic Information Systems Research. <http://dx.doi.org/10.1080/089419202014888791> [Internet] Taylor & Francis Group ; 2014 [cited 2022 Sep 30];27(5):557–571. [doi: 10.1080/08941920.2014.888791]
 11. Simons M, Price N, Kimble R, Tyack Z. Patient experiences of burn scars in adults and children and development of a health-related quality of life conceptual model: A qualitative study. *Burns Elsevier Ltd*; 2016. p. 620–632. PMID:26803365
 12. Simons M, Lim PCC, Kimble RM, Tyack Z. Towards a clinical and empirical definition of burn scarring: A template analysis using qualitative data. *Burns Elsevier*; 2018 Nov 1;44(7):1811–1819. PMID:30060903
 13. Jones LL, Calvert M, Moiemien N, Deeks JJ, Bishop J, Kinghorn P, Mathers J. Outcomes important to burns patients during scar management and how they compare to the concepts captured in burn-specific patient reported outcome measures. *Burns Elsevier Ltd*; 2017 Dec 1;43(8):1682–1692. PMID:29031889
 14. Li Y, Zhou L, Tang L, Liu M, Ming X, Shen F, Cui J, Meng X, Zhao J. Burn patients' experience of pain management: A qualitative study. *Burns Elsevier*; 2012 Mar 1;38(2):180–186. PMID:22079543
 15. Van Baar ME, Essink-Bot ML, Oen IMM, Dokter J, Boxma H, Van Beeck EF. Functional outcome after burns: A review. *Burns Elsevier*; 2006 Feb 1;32(1):1–9. PMID:16376020
 16. Stockly OR, Wolfe AE, Espinoza LF, Simko LC, Kowalske K, Carrougher GJ, Gibran N, Bamer AM, Meyer W, Rosenberg M, Rosenberg L, Kazis LE, Ryan CM, Schneider JC. The impact of electrical injuries on long-term outcomes: A Burn Model System National Database study. *Burns Elsevier*; 2020 Mar 1;46(2):352–359. PMID:31420267
 17. Karimi H, Momeni M, Vasigh M. Long term outcome and follow up of electrical injury. *J Acute Dis* No longer published by Elsevier; 2015 Jun 1;4(2):107–111. [doi: 10.1016/S2221-6189(15)30018-4]
 18. Pham TN, Kramer CB, Wang J, Rivara FP, Heimbach DM, Gibran NS, Klein MB. Epidemiology and outcomes of older adults with burn Injury: An analysis of the national burn repository. *J Burn Care Res* [Internet] J Burn Care Res; 2009 Jan [cited 2021 Jun 20];30(1):30–36. PMID:19060727
 19. Ryan CM, Lee A, Kazis LE, Schneider JC, Shapiro GD, Sheridan RL, Meyer WJ, Palmieri T, Pidcock FS, Reilly D, Tompkins RG. Recovery trajectories after burn injury in young adults: Does burn size matter? *J Burn Care Res* [Internet] Lippincott Williams and Wilkins; 2015 Jan 21 [cited 2021 Jun 20];36(1):118–129. PMID:25501787

20. Mason ST, Esselman P, Fraser R, Schomer K, Truitt A, Johnson K. Return to work after burn injury: A systematic review [Internet]. *J Burn Care Res. Oxford Academic*; 2012 [cited 2021 Jun 20]. p. 101–109. PMID:22138806
21. Cucuzzo NA, Ferrando A, Herndon DN. The Effects of Exercise Programming vs Traditional Outpatient Therapy in the Rehabilitation of Severely Burned Children. *J Burn Care Rehabil [Internet] Oxford Academic*; 2001 May 1 [cited 2022 Sep 27];22(3):214–220. PMID:11403243
22. Druery M, Brown TLH, Muller M. Long term functional outcomes and quality of life following severe burn injury. *Burns Elsevier*; 2005 Sep 1;31(6):692–695. PMID:16129223
23. Stull DE, Leidy NK, Parasuraman B, Chassany O. Optimal recall periods for patient-reported outcomes: challenges and potential solutions. <https://doi.org/10.1185/03007990902774765> [Internet] *Taylor & Francis*; 2009 Apr [cited 2022 Sep 27];25(4):929–942. PMID:19257798
24. Dolan AR, Goldberg EB, Cannuscio CC, Abrams MP, Feuerstein-Simon R, Marti XL, Mazique J, Schapira MM, Meisel ZF. Patient Perceptions About Opioid Risk Communications Within the Context of a Randomized Clinical Trial. *JAMA Netw Open [Internet] American Medical Association*; 2022 Aug 1 [cited 2022 Sep 27];5(8):e2227650–e2227650. PMID:35980634
25. Sebele-Mpofu FY. Saturation controversy in qualitative research: Complexities and underlying assumptions. A literature review. <http://www.editorialmanager.com/cogentsocsci> [Internet] *Cogent*; 2020 Jan 1 [cited 2022 May 12];6(1). [doi: 10.1080/23311886.2020.1838706]
26. Dey I. *Grounding grounded theory : guidelines for qualitative inquiry*. San Diego: Academic Press; 1999. ISBN:9780122146404, 0122146409
27. Hennink M, Kaiser BN. Sample sizes for saturation in qualitative research: A systematic review of empirical tests. *Soc Sci Med Pergamon*; 2022 Jan 1;292:114523. PMID:34785096
28. Kornhaber RA, De Jong AEE, McLean L. Rigorous, robust and systematic: Qualitative research and its contribution to burn care. An integrative review. *Burns Elsevier*; 2015 Dec 1;41(8):1619–1626. [doi: 10.1016/J.BURNS.2015.04.007]
29. Young A, Brookes S, Rumsey N, Blazeby J. Agreement on what to measure in randomised controlled trials in burn care: Study protocol for the development of a core outcome set. *BMJ Open BMJ Publishing Group*; 2017 Jun 1;7(6):e017267. [doi: 10.1136/bmjopen-2017-017267]
30. Williamson PR, Altman DG, Blazeby JM, Clarke M, Devane D, Gargon E, Tugwell P. Developing core outcome sets for clinical trials: Issues to consider. *Trials [Internet] BioMed Central*; 2012 Aug 6 [cited 2021 Jun 21];13(1):1–8. PMID:22867278
31. Mathers J, Moiemmen N, Bamford A, Gardiner F, Tarver J. Ensuring that the outcome domains proposed for use in burns research are relevant to adult burn patients: A systematic review of qualitative research evidence. *Burn Trauma [Internet] 2021 [cited 2021 Jun 21];8:30*. [doi: 10.1093/BURNST/TKAA030]
32. Auerbach SM, Martelli MF, Mercuri LG. Anxiety, information, interpersonal impacts, and adjustment to a stressful health care situation. *J Pers Soc Psychol [Internet] 1983 Jun [cited 2021 Nov 4];44(6):1284–1296*. [doi: 10.1037/0022-3514.44.6.1284]
33. Andrew JM. Recovery from surgery, with and without preparatory instruction, for three coping styles. *J Pers Soc Psychol [Internet] 1970 Jul [cited 2021 Nov 4];15(3):223–226*. [doi: 10.1037/H0029442]
34. Gammon J, Mulholland CW. Effect of preparatory information prior to elective total hip replacement on post-operative physical coping outcomes. *Int J Nurs Stud Pergamon*; 1996 Dec 1;33(6):589–604. [doi: 10.1016/S0020-7489(96)00019-3]